# Endogenous opioids in the nucleus accumbens promote approach to high-fat food in the absence of caloric need

**Kevin Caref[1], Saleem M Nicola[1,2]***

[1]Department of Neuroscience, Albert Einstein College of Medicine, Bronx, United States; [2]Department of Psychiatry, Albert Einstein College of Medicine, Bronx, United States

**Abstract** When relatively sated, people (and rodents) are still easily tempted to consume calorie-dense foods, particularly those containing fat and sugar. Consumption of such foods while calorically replete likely contributes to obesity. The nucleus accumbens (NAc) opioid system has long been viewed as a critical substrate for this behavior, mainly via contributions to the neural control of consumption and palatability. Here, we test the hypothesis that endogenous NAc opioids also promote appetitive approach to calorie-dense food in states of relatively high satiety. We simultaneously recorded NAc neuronal firing and infused a μ-opioid receptor antagonist into the NAc while rats performed a cued approach task in which appetitive and consummatory phases were well separated. The results reveal elements of a neural mechanism by which NAc opioids promote approach to high-fat food despite the lack of caloric need, demonstrating a potential means by which the brain is biased towards overconsumption of palatable food.
DOI: https://doi.org/10.7554/eLife.34955.001

*For correspondence:
saleem.nicola@einstein.yu.edu

Competing interests: The authors declare that no competing interests exist.

## Introduction

People often seek and consume calorie-dense food in the absence of hunger, and this behavior has profound implications for human health. Although preference for sweet and fatty foods may once have been adaptive, it now very likely contributes to epidemic rates of obesity and diabetes. Thus, understanding the neural mechanisms that guide seeking of highly palatable foods is an important step in the search for novel therapies that could combat these diseases by reducing caloric intake. One candidate neural substrate is the brain's opioid system, particularly in the ventral striatum. A role for this circuitry is supported by observations that the ventral striatum, and in particular the nucleus accumbens (NAc), is richly endowed with both opioid peptides and their respective receptors (*Mansour et al., 1988*), and that activation of NAc μ-opioid receptors (MORs) selectively augments consumption of palatable food (*Bakshi and Kelley, 1993*; *Mucha and Iversen, 1986*; *Zhang et al., 1998*; *Zhang and Kelley, 1997*) and of preferred flavors (*Woolley et al., 2006*). Moreover, activation of NAc MORs increases hedonic taste reactions to palatable food (*Peciña and Berridge, 2000*). Thus, opioids are thought to contribute primarily to the encoding of hedonic responses to food, which in turn reinforces the assignment of incentive salience to cues associated with palatable reward (*Berridge, 2009*; *Castro and Berridge, 2014*).

However, several observations indicate that this view is incomplete. First, blockade of NAc MORs does not consistently reduce calorie-dense food consumption (*Bodnar et al., 1995*; *Kelley et al., 1996*; *Lardeux et al., 2015*; *MacDonald et al., 2003*; *Ward et al., 2006*). In addition, activation of NAc MORs increases certain measures of reward-seeking behavior, including breaking point on a progressive-ratio task (*Zhang et al., 2003*) and lever pressing in the presence of food-predictive cues in a Pavlovian-to-instrumental transfer task (*Peciña and Berridge, 2013*). These studies suggest

**eLife digest** Imagine that you have just finished Thanksgiving dinner. You are completely full, having eaten large portions of turkey, green beans and mashed potatoes. Yet, despite feeling full, you still find yourself tempted by a slice of pie for dessert, maybe even with ice cream on top. Why is it that in such a state of fullness, you desire a slice of pie but not, say, another helping of green beans?

The answer may lie in the way the brain responds to food when we do not need any more calories. At such times, your brain drives you to continue eating only those foods that are tasty and calorie-dense. This preference for fatty and sweet foods may have been helpful in the past when we could not be certain where our next meal would come from. But in modern times, the widespread availability of food makes this preference potentially harmful. For example, the drive to consume fatty and sweet foods even when not hungry may now be contributing to soaring levels of obesity and type 2 diabetes.

What exactly is happening inside the brain to produce this behavior? Previous work has implicated a structure called the nucleus accumbens. When scientists activated proteins called mu opioid receptors within the nucleus accumbens, animals ate more of the foods that they find tasty. However, they were not as interested in eating more of the foods that they are more ambivalent towards.

Caref and Nicola now show that preventing opioid binding makes rats unwilling to respond to a cue to obtain cream, an appetizing, high-fat reward. It also abolishes the brain activity that drives the rats to respond the cue. Crucially, however, this effect only occurs in rats that are not hungry.

It therefore appears that opioid binding in the nucleus accumbens drives animals to approach and eat high-fat foods, but only when they do not need the calories. That is, it increases fat consumption in animals that are not actually hungry. A drug that selectively blocks mu opioid receptors in the nucleus accumbens may reduce this behavior. Such a drug could potentially help to prevent obesity and the health problems associated with it.

DOI: https://doi.org/10.7554/eLife.34955.002

that NAc MOR activation could promote food-seeking behavior directly, instead of (or in addition to) doing so by enhancing the hedonic or reinforcing effects of the food. Consequently, we hypothesize that when people or animals are sated, their preferences shift toward palatable food because endogenous ligands of NAc MORs selectively promote seeking of calorie-dense foods. This idea has not yet been tested because few studies have examined the contribution of NAc MORs activated by endogenous ligands to appetitive (food-seeking) as opposed to consummatory behaviors.

Here, we address this gap in our knowledge by using a conditioned-stimulus (CS) task that disambiguates appetitive from consummatory behavior. In this task, rats perform an approach response to a reward-predictive cue to obtain a highly palatable, calorie-dense liquid food (cream). NAc neurons encode both cued approach and reward consumption phases of such behaviors (*Ambroggi et al., 2011*; *du Hoffmann and Nicola, 2014*; *McGinty et al., 2013*; *Morrison et al., 2017*; *Nicola, 2010*; *Nicola et al., 2004a*; *2004b*; *Taha and Fields, 2005*), and cue-evoked excitations are necessary for the approach response (*Ambroggi et al., 2008*; *du Hoffmann and Nicola, 2014*; *Yun et al., 2004*). By simultaneously recording from NAc neurons and injecting a MOR antagonist into the NAc, we show that NAc MOR activation is required for both behavioral responding to reward-predictive cues and the neural encoding of those cues by NAc neurons. Importantly, these effects were observed in ad libitum chow-fed rats but not in those that had been food restricted. This striking dichotomy indicates that activation of NAc MORs promotes the approach to palatable food only in the absence of a homeostatic need for calories – i.e., hunger – suggesting that these receptors contribute to a neural mechanism that drives intake of calorie-dense food specifically in the state of satiety.

## Results

### NAc MOR activation is required for conditioned approach in ad-libitum fed, but not food-restricted rats

Free-fed rats were trained on a CS task (*Figure 1A*) in which they were presented with an unpredictable series of two auditory tones with a mean intertrial interval (ITI) of 30 s. The CS+ tone was reward predictive, such that rats could earn a droplet of heavy cream by making a head entry into the reward receptacle during the 5 s cue presentation. The 5 s CS- tone was not reward predictive and receptacle responses during this cue or the ITI were recorded but had no programmed consequence. CS task performance was assessed by computing a response ratio, defined as the percentage of cue presentations of a particular type (either CS+ or CS-) that the animal responded to. Once rats learned to discriminate between cues (see Materials and methods), they were implanted bilaterally with cannulae targeting the NAc core (see *Figure 1—figure supplement 2A* for histological examination of injection sites). Following recovery from surgery, rats underwent several retraining sessions before the start of the experiment, and a subset of rats was food restricted for at least 7 days concurrent with retraining.

By the final day of retraining, food-restricted rats responded to a significantly higher proportion of cues than free-fed rats (*Figure 1B*, see figure legend for statistics), with free-fed rats exhibiting a pronounced decline in responding over the course of the session (*Figure 1C*, solid black line). Rats were then bilaterally injected every other day with the selective MOR antagonist CTAP (0, 2, or 4 µg/side; *Figure 1C*) prior to the session. Bilateral CTAP injection significantly attenuated responding to the CS+ in free-fed rats (*Figure 1C*, blue solid lines) but, strikingly, had no effect in food-restricted rats (*Figure 1C*, blue dashed lines), suggesting a state-dependent contribution of MOR activation to reward-seeking behavior. While food-restricted rats had higher levels of CS- responding than free-fed rats, CTAP had no effect on CS- response ratio in either group (*Figure 1—figure supplement 1*).

Because there is evidence that NAc MOR activation selectively enhances consumption of fat in lieu of carbohydrates (*Katsuura et al., 2011*; *Zhang et al., 1998*), we next asked whether the CTAP effect could also be observed in free-fed rats performing the same task for 3% liquid sucrose reward. Interestingly, while the pattern of CS+ responding for sucrose was similar to responding for cream (solid black lines in *Figure 1D and C*, respectively), CTAP had no effect on responding for sucrose (*Figure 1D*). Taken together, these data suggest that MOR blockade preferentially affects responding to fat-predictive cues, and that this effect cannot be attributed to interference with more general motivational or arousal-related neural processes.

### Neural encoding of reward-predictive cues by NAc neurons is different in free-fed and food-restricted rats

We next sought to understand the neural mechanism by which CTAP attenuated behavioral responding to reward-predictive cues in free-fed, but not food-restricted rats. Because previous studies have demonstrated that many NAc neurons are excited by reward-predictive cues (*Ambroggi et al., 2011*; *Ambroggi et al., 2008*; *du Hoffmann and Nicola, 2014*; *McGinty et al., 2013*; *Nicola et al., 2004a*; *Yun et al., 2004*), and further, that these cue-evoked excitations are required for behavioral responding to those cues (*du Hoffmann and Nicola, 2014*; *Yun et al., 2004*), we hypothesized that NAc cue-evoked excitations serve as the neural effector of MOR activation. To address whether this is the case, we first recorded from NAc neurons in both free-fed and food-restricted rats during performance of the CS task. Because free-fed rats respond to a lower proportion of cues, and because cue-excited neurons fired much more in trials in which the rat responded than when it did not (*Figure 2—figure supplement 1A,B*), we constrained this analysis to trials in which rats responded to the CS+. Out of 83 neurons recorded in 12 free-fed rats, 45 neurons were excited by the CS+ (54.2%; *Figure 2A*), as opposed to 91 out of 122 neurons (74.5%; *Figure 2B*) recorded from five food-restricted rats. Further, the magnitude of the cue-excited population response in food-restricted rats was significantly greater than the response in free-fed rats (*Figure 2C,D*), as was the fraction of 50 ms time bins after cue onset with significant excitations (*Figure 2E,F*, upper traces/dots).

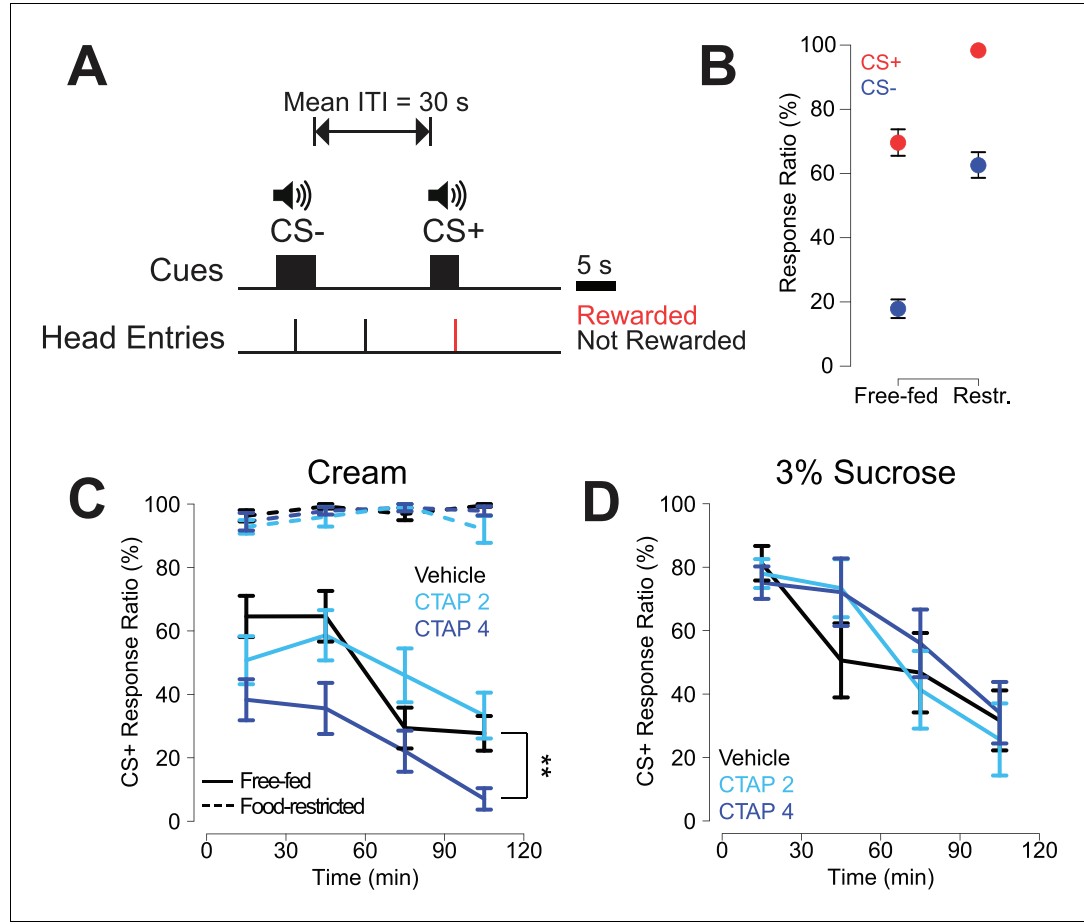

**Figure 1.** CTAP-induced impairment of cued approach behavior is observed in free-fed animals receiving cream reward, but not in food-restricted animals or those receiving sucrose reward. (**A**) CS task diagram. Only head entries into the receptacle during CS+ presentation are rewarded, at which point the cue is terminated. (**B**) Response ratios for each cue type on the final day of retraining before the microinjection experiment began. For each cue type, response ratio is defined as the proportion of cues responded to out of the number of cues presented. Food-restricted rats (N = 7, right) responded more to both the CS+ (red dots) and the CS- (blue dots) than free-fed rats (N = 16, left); ***p<0.001, Wilcoxon. (**C**) Bilateral CTAP (4 µg/side) injection into the NAc reduced CS+ response ratio in free-fed (solid lines), but not food-restricted rats (dashed lines). A two-factor ANOVA (time x dose) performed on free-fed rats revealed significant main effects of dose ($F_{2,180}$ = 12.28, p<0.001) and time ($F_{3,180}$ = 12.28, p<0.001). The interaction between time x dose was not significant ($F_{6,180}$ = 1.06, p=0.39). All points represent mean ± SEM. N = 16 for free-fed rats; N = 7 for food-restricted rats. **p<0.01 over the whole session for the 4 µg dose vs vehicle, Holm-Sidak adjusted. ANOVA revealed no significant effects in food-restricted rats (dose, $F_{2,64}$ = 2.27, p=0.11; time, $F_{3,64}$ = 2.11, p=0.11; dose x time $F_{6,64}$ = 1.16, p=0.34). (**D**) When the reward was 3% sucrose instead of heavy cream, CTAP had no effect on CS+ responding in free-fed rats (N = 8). A two-factor ANOVA revealed a main effect of time ($F_{3,84}$ = 13.54, p<0.001), but not dose ($F_{2,84}$ = 0.51, p=0.605). The time x dose interaction was not significant ($F_{6,84}$ = 0.70, p=0.66).

DOI: https://doi.org/10.7554/eLife.34955.003

The following figure supplements are available for figure 1:

**Figure supplement 1.** CS- response ratio for CTAP injections.

DOI: https://doi.org/10.7554/eLife.34955.004

**Figure supplement 2.** Histological examination of probe locations.

DOI: https://doi.org/10.7554/eLife.34955.005

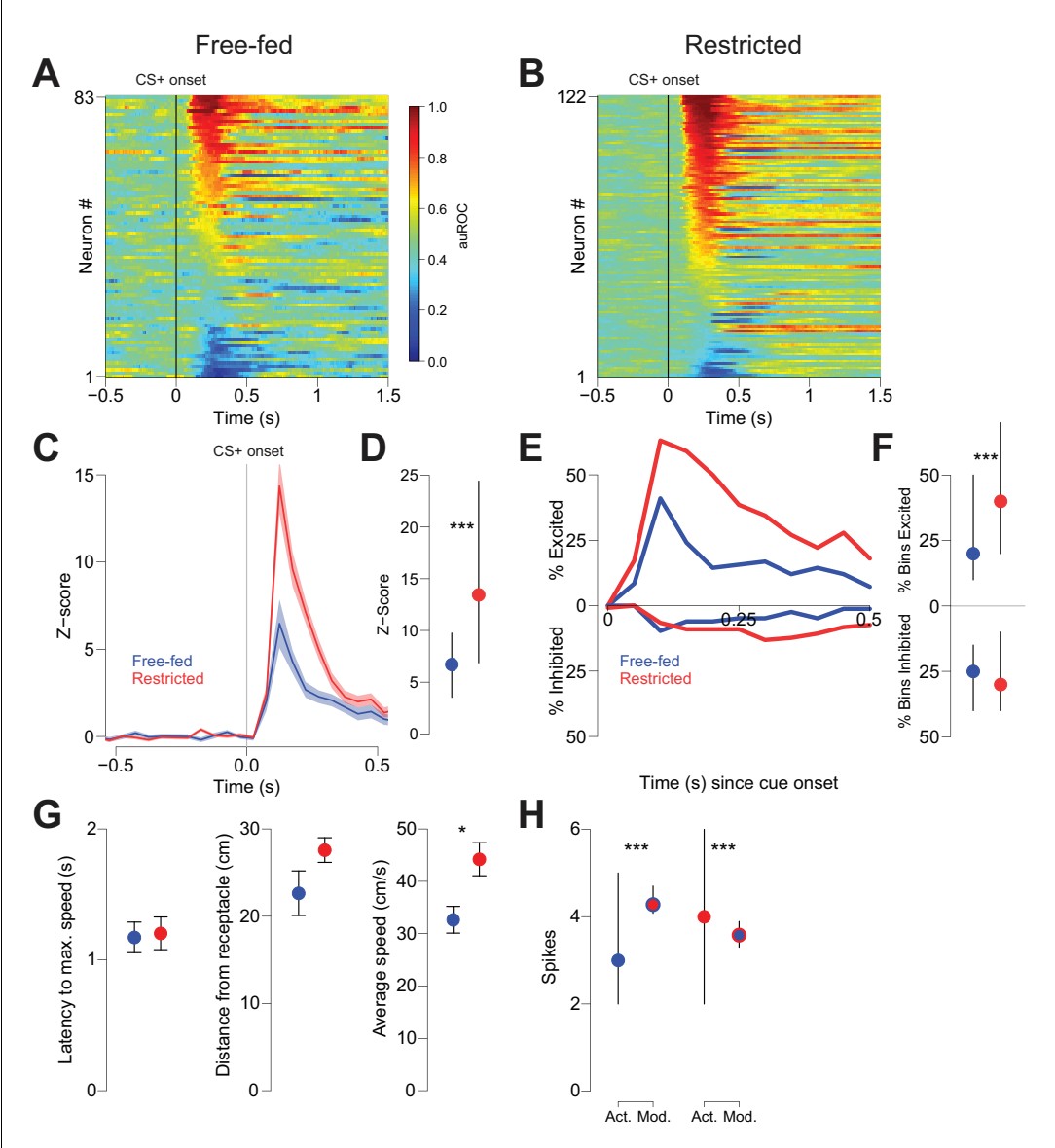

**Figure 2.** Cue-evoked excitations, but not inhibitions, are more robust in food-restricted than free-fed rats. (A) Heat map showing CS+ evoked activity of all neurons recorded in free-fed rats. Each row represents an individual neuron's response to the CS+ averaged across all rewarded trials during the session. Neurons are sorted by the intensity of the response between 100–300 ms post-cue. Colors indicate area under the ROC curve (auROC) comparing firing in the time bin to baseline; hotter colors represent excitation (auROC > 0.5), cooler colors represent inhibition (auROC < 0.5). Data are smoothed for display purposes. (B) Same as A), but for food-restricted rats. (C) Mean perievent time histogram for all neurons classified as cue-excited in A) and B). Only trials with a behavioral response were included. Lines represent the mean response; shaded regions represent ± SEM. (D) The population response of neurons from food-restricted rats (red) is significantly greater than the response in free-fed rats (blue) in the window between 100–300 ms post-cue. Dots indicate the median response, lines indicate inter-quartile range; ***p<0.001, Wilcoxon. (E) Percentage of significantly excited (upper traces) or inhibited (lower traces) neurons in each 50 ms bin compared to pre-cue baseline from the entire free-fed (blue) or restricted (red) populations. (F) Proportion of post-cue bins with significant excitation or inhibition for each cue-excited or cue-inhibited neuron in the 500 ms after cue onset. Dots indicate median proportion of excited bins, lines indicate inter-quartile range; ***p<0.001, Wilcoxon. Only neurons classified as either cue-excited or -inhibited were included (see 'Analysis of neural data' section of Materials and methods for the criteria used for neuronal classification). (G) For the subset of sessions with video tracking available (N = 4 rats over five sessions for free-fed, N = 3 rats over five sessions for restricted), mean ± SEM of latency to maximum speed after CS+ onset (left), distance from the receptacle at CS+ onset (center), and average speed after CS + onset (right) for free-fed (blue) and restricted (red) sessions; *p<0.05, Wilcoxon. (H) Left, using the same sessions as in G), the median and inter-quartile range of the actual spike count from sated sessions (blue dot) on trials in which the rat responded to the CS+ compared to the modeled spike count computed using regression coefficients from the food-restricted sessions and the behavioral parameters from free-fed sessions (red dot with blue

*Figure 2 continued*

outline). Right, the spike counts from food-restricted sessions (red dot) compared to the modeled spike count computed using regression coefficients from the free-fed population and behavioral parameters from the food-restricted population (blue dot with red outline); ***p<0.001, Wilcoxon.

DOI: https://doi.org/10.7554/eLife.34955.006

The following figure supplements are available for figure 2:

**Figure supplement 1.** GLM Validation.

DOI: https://doi.org/10.7554/eLife.34955.007

**Figure supplement 2.** Distributions of baseline firing rates of neurons recorded from free-fed (blue bars) and food-restricted rats (red bars).

DOI: https://doi.org/10.7554/eLife.34955.008

In addition to cue-evoked excitations, we also observed smaller populations of cue-inhibited neurons in both free-fed and food-restricted rats. In free-fed rats, 16.8% (14 out of 83) of neurons were inhibited by the CS+, compared to 18.0% (22 out of 122) in food-restricted rats (*Figure 2A,B*). Unlike with excitations, there was no difference in the fraction of significantly inhibited bins (*Figure 2F*, lower dots). Moreover, there was no difference in the baseline firing rate between the two populations (*Figure 2—figure supplement 2*).

Because cue-evoked excitations have been shown to encode certain spatial and behavioral elements of response vigor such as distance from receptacle at cue onset, latency to maximum speed, and average speed (*McGinty et al., 2013*; *Morrison et al., 2017*), we reasoned that the observed difference in the magnitude of excitation between free-fed and food-restricted populations could be explained by differences in either response vigor or the encoding of response vigor. To determine if this was the case, we first compared the behavioral metrics of restricted and free-fed rats. We found that the average speed of approach after cue onset was significantly greater in restricted rats, while latency to maximum speed and distance did not differ (*Figure 2G*). To determine if this difference could account for the observed difference in cue-evoked excitation, we used a generalized linear model (GLM) to regress the post-cue spike count of each population (restricted and free fed) of cue-excited neurons against the three behavioral parameters (*Figure 2—figure supplement 1C–G*). Next, we used the GLM and coefficients generated from neurons recorded in food-restricted rats to model the spike count on each trial obtained in free-fed rats by entering each trial's behavioral parameters into the regression equation. Finally, we performed the same analysis, but instead used the GLM generated from free-fed rats to model the spike counts in restricted rats.

To interpret the results, we reasoned that if the difference in the magnitude of excitation between restricted and free-fed animals were wholly accounted for by the differences in behavior, then there would not be a significant difference between the modeled spike counts (using regression coefficients from the other population) and the actual spike counts from that population. In fact, we observed significant differences in both analyses: when we compared the actual spike counts from free-fed rats to the spike counts predicted by the GLM obtained from food-restricted rats, the modeled spike counts were significantly higher (*Figure 2H*, left panel). Similarly, spike counts that were modeled with the GLM obtained from free-fed rats were significantly lower than the actual spike counts from the restricted rats (*Figure 2H*, right panel). We then employed *Equation 2* (See *Generalized Linear Model (GLM) fitting* in Materials and methods) to test whether the GLMs obtained from each population were statistically distinct, or whether they model the same overall neural population. In brief, this test compares the pooled residual deviance from the two GLMs to the residual deviance of a GLM containing all data from both populations while accounting for the number of regressors. Consistent with the modeling analysis from *Figure 2H*, the two GLMs do in fact model separate populations, and not the same overall population ($F_{4,2634} = 10.97$; p<0.001). Taken together, these results indicate that lower cue-evoked excitation in free-fed than in restricted rats is not wholly accounted for by lower response vigor, which suggests that additional, unaccounted factors push firing rate lower in free-fed animals than their lower response vigor would predict (or, equivalently, higher in restricted animals than their greater vigor would predict).

Many NAc neurons were modulated during reward consumption (*Figure 3*). To determine whether neural activity during this epoch differed based on caloric need, we considered firing in the first 3 s following each initial rewarded receptacle entry. Although some excitations and inhibitions lasted longer than 3 s, receptacle exit almost always occurred later than this time point (*Figure 8—figure supplement 1A,B*), assuring that our analysis window included only periods when the animal

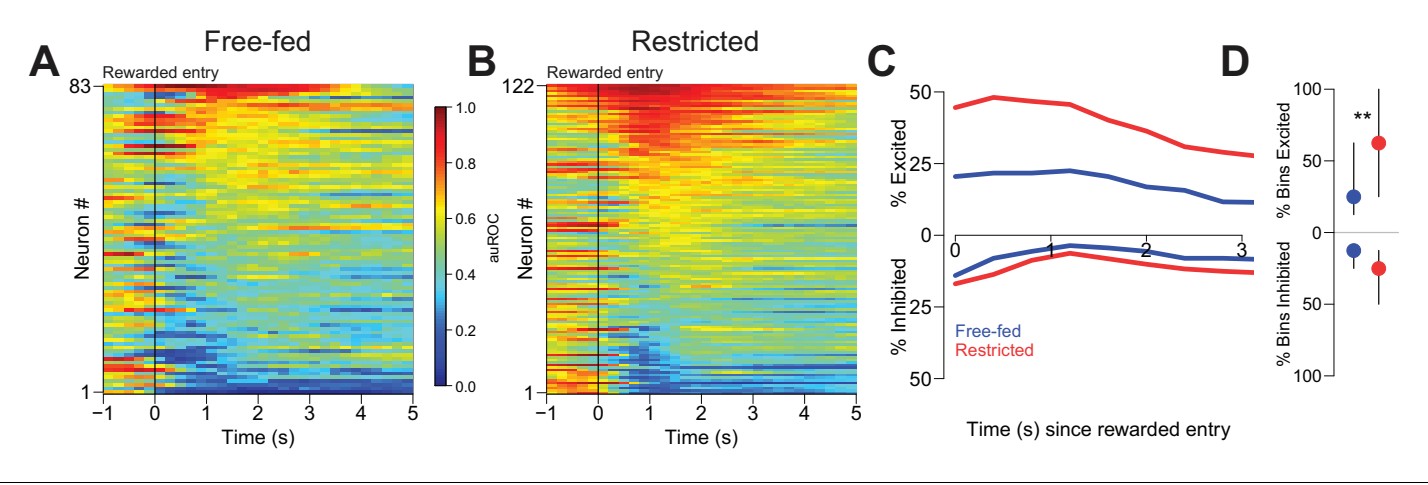

**Figure 3.** Reward-associated excitations, but not inhibitions, are different in free-fed and restricted populations. (A) Heat map time-locked to the receptacle entry that triggers reward delivery in free-fed rats. The baseline period is the pre-cue epoch for each rewarded trial. auROC values are computed in an identical fashion as in *Figure 2*, except here the bin size is 200 ms. Neurons are sorted by the magnitude of their response between 1–3 s after rewarded-entry. (B) Same as A), but for food-restricted rats. (C) Proportion of significantly excited (upper traces) and inhibited (lower traces) neurons for each population in each 400 ms bin during the 3 s epoch after rewarded entry compared to pre-cue baseline. (D) For each significantly excited (upper dots) or inhibited (lower dots) neuron, the percentage of bins with significant excitation or inhibition. Dots represent the median proportion of excited or inhibited bins, lines represent the inter-quartile range; **p<0.01, Wilcoxon.
DOI: https://doi.org/10.7554/eLife.34955.009

was in the receptacle. In free-fed rats, 37.3% (31 out of 83) of neurons were excited for at least a single 400 ms bin following entry into the receptacle on rewarded trials, compared to 60.7% (74 out 122) of neurons in food-restricted rats; moreover, more bins exhibited significant excitation in restricted than free-fed rats (*Figure 3D*, upper dots). There was also a prominent population of reward-associated inhibitions in each group: 32.5% (27 out of 83) were inhibited for at least one bin following rewarded receptacle entry in free-fed rats, compared to 34.4% (42 out of 122) in food-restricted rats. Unlike excitations, there was no difference in the fraction of bins with significant inhibition between the two groups (*Figure 3D*, lower dots).

## CTAP attenuates the encoding of cues by NAc neurons in free-fed, but not food-restricted rats

It has been demonstrated previously that the magnitude of cue-evoked excitation predicts the vigor of the subsequent cued approach response (*du Hoffmann and Nicola, 2014*; *McGinty et al., 2013*) and further, that these excitations are required for the behavior (*du Hoffmann and Nicola, 2014*; *Yun et al., 2004*). Therefore, we hypothesized that activation of MORs facilitates cue-evoked excitations in free-fed rats, but not in food-restricted rats. This would explain why CTAP injection impaired cued approach behavior in free-fed rats but not restricted rats (*Figure 1*). To test this hypothesis, we injected CTAP into the NAc while simultaneously recording NAc unit activity. Rats trained on the CS task with cream reward were implanted bilaterally with circular microelectrode arrays surrounding a central microinjection guide cannula (see *Figure 1—figure supplement 2B* for histological examination of injection sites). These arrays allow for the injection of a drug into the same brain region from which neural recordings are being obtained, thereby enabling within-session comparisons of pre- vs post-injection behavior and neural activity (*du Hoffmann et al., 2011*; *du Hoffmann and Nicola, 2014*). Rats performed the CS task for a 33 min baseline period, after which CTAP was injected by remote activation of a syringe pump (i.e., without interrupting the ongoing behavior). The pre-injection baseline behavioral performance and neural activity was then compared to the 33 min window after drug injection. In a subset of subjects, rats' positions in the operant chamber were tracked via two LEDs mounted on the neural recording headstage.

As we previously observed (*Figure 1C*), in bilaterally-injected free-fed rats, CTAP sharply attenuated responding to the CS+ (*Figure 4A,B* blue trace and bars), while the drug had no effect in food-

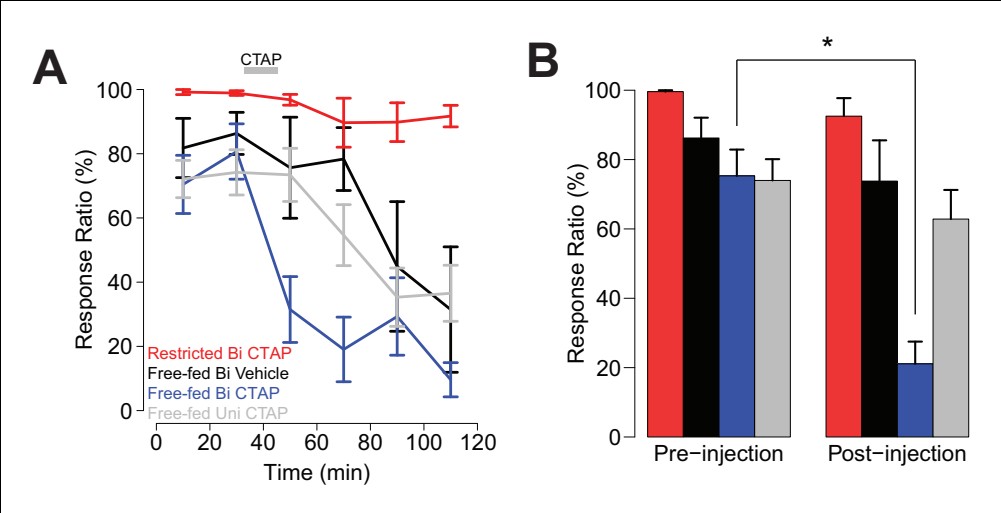

**Figure 4.** Bilateral, but not unilateral CTAP injections delivered mid-session impair cued approach behavior. Colors represent the same groups in both panels. (**A**) Crunch plot showing mean CS+ response ratio in four different groups of animals. Vehicle: N = 5 rats over five sessions; free-fed bilateral CTAP: N = 5 rats over eight sessions; restricted bilateral CTAP: N = 5 rats over nine sessions; free-fed unilateral CTAP: N = 6 rats over 11 sessions. Bin size = 20 min. Gray bar indicates time course of CTAP (or vehicle) injection. A three-factor ANOVA (drug x time x satiety state) revealed significant main effects of drug ($F_{1,130}$ = 23.06, p<0.001), time ($F_{5,126}$ = 10.20, p<0.001), and satiety state ($F_{1,130}$ = 112.13, p<0.001), as well as significant interactions between drug x time ($F_{5,126}$ = 2.43, p=0.039) and satiety state x time ($F_{5,126}$ = 4.21, p=0.002). (**B**) Direct comparison of pre- vs. post-injection epochs for CS+ response ratio. Colors are the same as in **A**). Only free-fed rats receiving bilateral CTAP injections demonstrated a significant decrease in responding post-injection (*p<0.05, Wilcoxon, Holm-Sidak corrected).
DOI: https://doi.org/10.7554/eLife.34955.010

The following figure supplement is available for figure 4:

**Figure supplement 1.** CTAP increases the latency to initiate locomotion following cue onset.
DOI: https://doi.org/10.7554/eLife.34955.011

restricted rats (*Figure 4A,B* red trace and bars). In contrast, both unilaterally-injected CTAP and saline-injected rats (*Figure 4A*, gray and black traces, respectively) exhibited a slower decline in responding over the session, consistent with the rate of decline previously observed in free-fed rats (black traces in *Figure 1B,D*) (*du Hoffmann and Nicola, 2016*). These slow declines in responding were accompanied by increases in latency to initiate movement, which were only slightly further increased after bilateral CTAP injection (*Figure 4—figure supplement 1A–C*). In addition, locomotor activity during the ITI was not further reduced by bilateral CTAP injection (*Figure 4—figure supplement 1D*). These results suggest that the CTAP-induced impairment of cued approach behavior (*Figure 4A,B*) was not due to generalized impairment of motor ability.

During the behavior shown in *Figure 4*, cue-evoked excitations in NAc neurons were significantly attenuated following CTAP injection in free-fed rats (*Figure 5A,B* and *Figure 5—figure supplement 1A*). This was true of both the magnitude of the excitations (*Figure 5E,F*) and the fraction of significantly excited bins (*Figure 5J*). In contrast, the magnitude of cue-evoked excitations in food-restricted animals was unchanged (*Figure 5C,D,G,H*), as was the fraction of significantly excited bins (*Figure 5L*). (There were insufficient cue-evoked inhibitions in free-fed rats to assess the drug's effects; for the six significantly inhibited neurons in food-restricted rats, a slight decrease in the fraction of inhibited bins pre- vs post-injection did not achieve statistical significance; p=0.06, Wilcoxon.)

To determine whether reductions in behavioral performance could have contributed to the CTAP-induced reduction of cue-evoked excitation, we examined firing during unilateral CTAP injections, which had no discernable behavioral effects in free-fed rats (*Figure 4*). To control for the possibility that during some sessions rats would respond to fewer cues post-injection (due to the gradual decline in cued approach responding in free-fed rats), we considered only trials in which rats responded to the CS+. In neurons ipsilateral to the CTAP injection (i.e., neurons directly exposed to

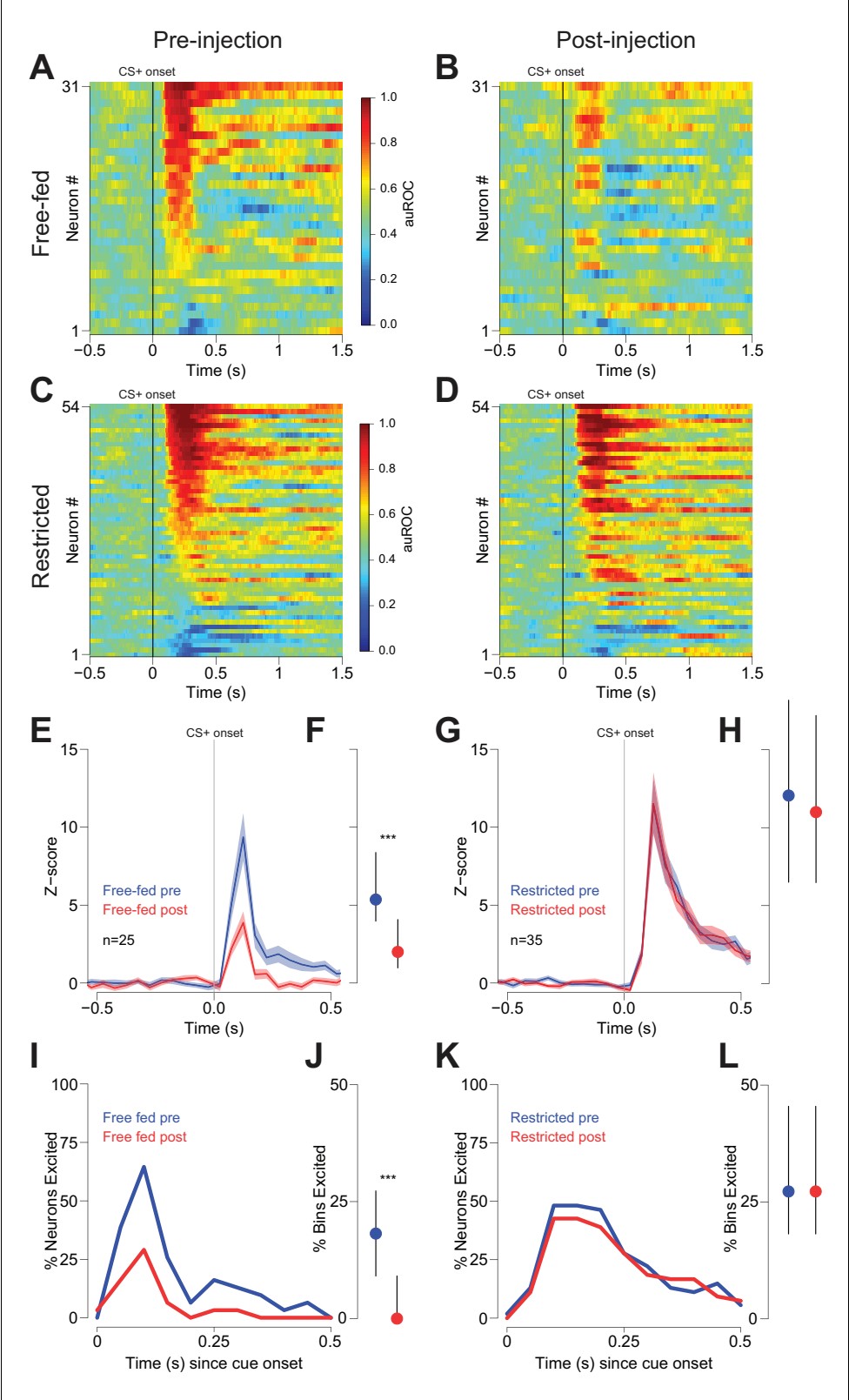

**Figure 5.** Cue-evoked excitations are reduced after bilateral CTAP injection in free-fed but not restricted rats. (A) and (B) Population heat maps for all neurons recorded from free-fed rats pre-injection (A) vs. post-injection (B). The sort order is the same for each pair of heat maps. (C) and (D) Same as (A and B), except for food-restricted rats. (E) Peri-stimulus time histogram (PSTH) of neurons classified as cue-excited for from the free-fed population. The blue trace represents mean z-score ± SEM for the pre-injection epoch; the red trace represents the post-injection epoch. (F) Dot plot indicating median

*Figure 5 continued on next page*

*Figure 5 continued*

and interquartile range of the response in each epoch of cue-excited neurons between 100–300 ms following cue onset. Colors are the same as in **E**). ***p<0.001, Wilcoxon. (**G**) and (**H**) Same as (**E** and **F**), but for food-restricted rats. (**I**) Proportions of cue-excited neurons in each 50 ms bin following cue onset for free-fed rats pre-injection (blue trace) vs post-injection (red trace). (**J**) Neurons from free-fed rats exhibited a significant post-injection decrease in the proportion of significantly excited bins. Only neurons with significant post-cue excitation were included in this analysis. ***p<0.001, Wilcoxon. (**K**) and (**L**) Same as **I**) and **J**), except for food-restricted rats, which did not exhibit a difference in the proportion of significantly excited bins pre- vs post-injection, p=0.55, Wilcoxon.

DOI: https://doi.org/10.7554/eLife.34955.012

The following figure supplements are available for figure 5:

**Figure supplement 1.** Raster plots of two representative cue-excited neurons.

DOI: https://doi.org/10.7554/eLife.34955.013

**Figure supplement 2.** Effect of CTAP on baseline firing rate.

DOI: https://doi.org/10.7554/eLife.34955.014

drug), we observed a significant reduction in the magnitude of cue-evoked excitations post-injection (*Figure 6A,B,E,F*). In contrast, neurons contralateral to the injection (i.e., not exposed to drug; *Figure 6C,D*) exhibited no significant reduction in the magnitude of cue-evoked excitations (*Figure 6G,H*). Additionally, among neurons that were classified as cue-excited or cue-inhibited, the proportion of bins with significant excitation or inhibition was significantly decreased for neurons ipsilateral to the injection (*Figure 6I,J*), but not for contralateral neurons (*Figure 6K,L*). Finally, saline injection had no effect on cue-evoked neural activity (*Figure 7*), demonstrating that the observed change in firing or behavior following CTAP injection cannot be attributed to any physical perturbation by the injection itself. These results suggest that in free-fed (but not restricted) rats, activation of MORs by endogenous ligands in the NAc is required for cue-evoked excitations that, in turn, drive approach to the reward receptacle.

## CTAP injection had no effect on baseline firing rate in free-fed rats

Because MORs are classically inhibitory, we tested the possibility that CTAP increases the baseline firing rate of NAc neurons, an effect that could theoretically contribute to the impairment of cued approach behavior. However, CTAP injection had no effect on baseline firing rate in free-fed rats, as the slope of the regression line of pre- vs. post- baseline firing rate did not significantly differ from the unity line (*Figure 5—figure supplement 2A*). Neurons from food-restricted rats demonstrated a slight reduction in baseline that may be attributable to the presence of outliers (*Figure 5—figure supplement 2B*), and which is unlikely to have affected behavior, as CTAP did not impact cued approach in restricted animals.

## CTAP injection does not affect consumption-related firing

Because the existing literature suggests that (1) the μ-opioid system in the NAc maintains hedonic responses to food (*Bakshi and Kelley, 1993*; *Bodnar et al., 1995*; *Kelley et al., 1996*; *Mucha and Iversen, 1986*; *Peciña and Berridge, 2000*; *Zhang et al., 1998*; *Zhang and Kelley, 1997*), and (2) a population of neurons in the NAc encodes relative palatability (*Taha and Fields, 2005*), we hypothesized that neuronal modulation during reward consumption might contribute to subsequent reinforcement of cued approach. Furthermore, we reasoned that CTAP might affect reinforcement and thus change the probability of cued approach behavior in free-fed animals by interfering with consumption-associated neural activity. We performed three analyses of our data to address this possibility. First, we reasoned that if firing during reward consumption contributed to reinforcement, then consumption-associated firing on a given trial should predict the probability of a behavioral response on the next trial. To test this idea, we used data from free-fed, uninjected rats to run a logistic regression to ask whether the number of spikes between 1–3 s after the rewarded entry on a given trial influenced the response probability on the subsequent trial. For both the consumption-excited and consumption-inhibited population, the spike count on a given trial did not significantly contribute to response probability (p=0.18 and p=0.20, respectively, Wald test), suggesting that reward-associated firing does not influence subsequent behavioral responding (at least on a trial-to-trial basis; we cannot rule out the possibility that firing on a given trial may influence responding at some later point in the session).

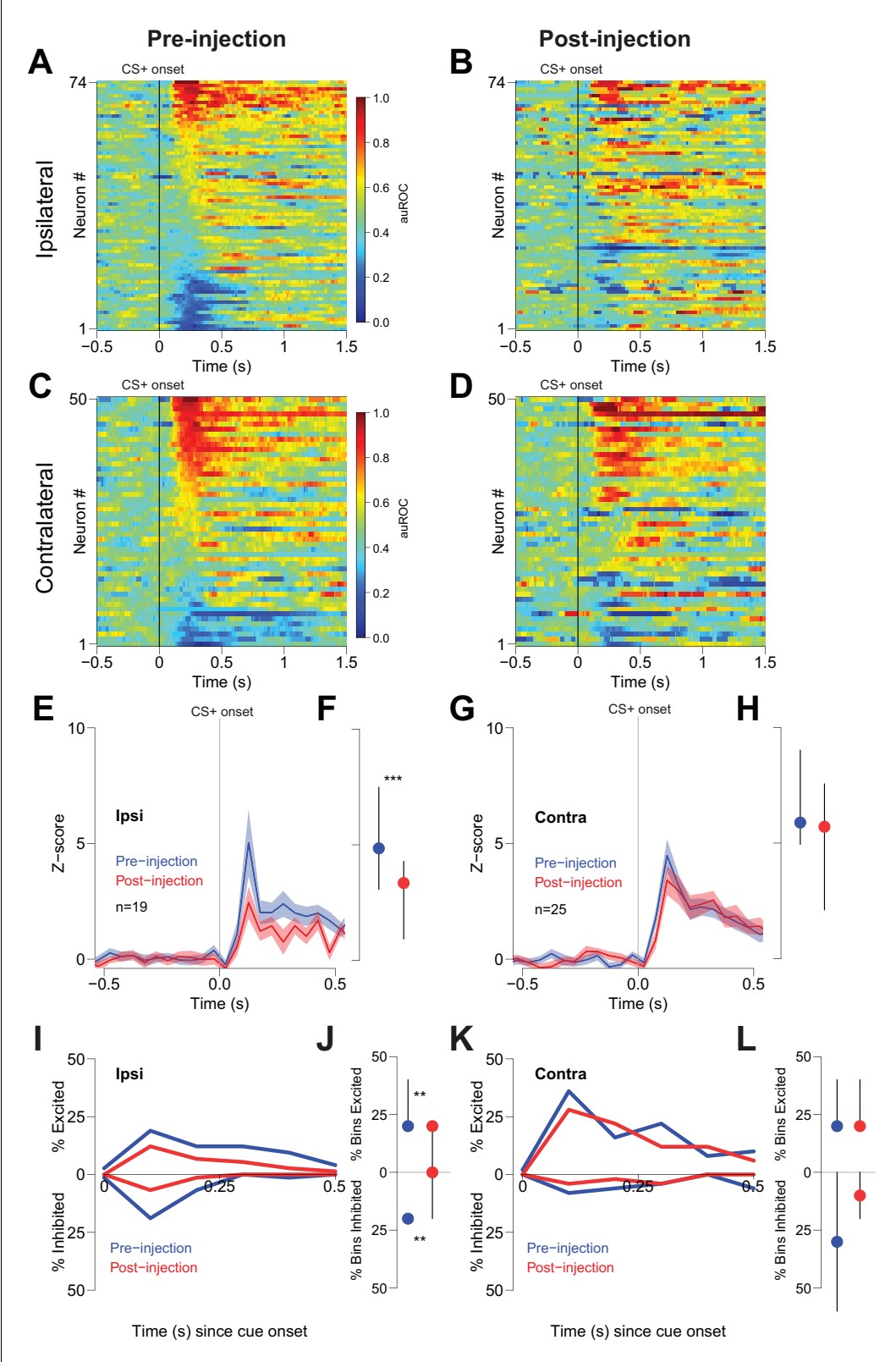

**Figure 6.** Cue-evoked excitations and inhibitions ipsilateral, but not contralateral, to CTAP injection are reduced. (**A**) Pre-injection population heat maps for neurons recorded ipsilateral to the site of CTAP injection. (**B**) Post-injection heat map corresponding to **A**); the sort order is the same for each heat map. (**C**) and (**D**) Same as (**A** and **B**), but for neurons recorded contralateral to the site of CTAP injection. (**E**) PSTH of neurons classified as cue-excited for the ipsilateral population. Blue trace represents mean z-score ± SEM for the pre-injection epoch; red trace represents the post-injection
*Figure 6 continued on next page*

*Figure 6 continued*

epoch. (F) Dot plot indicating median and interquartile range of the response in each epoch of cue-excited neurons between 100–300 ms following cue onset. ***p<0.001, Wilcoxon. (G) and (H) Same as (E and F), except for neurons contralateral to the injection. There was no significant change in the magnitude of the response pre- vs post-injection. p=0.19, Wilcoxon. (I) Proportions of cue-excited (upper traces) and cue-inhibited (lower traces) neurons in each 50 ms bin following cue onset pre-injection (blue traces) vs post-injection (red traces) out of all ipsilateral neurons. (J) Proportion of bins with significant excitation (upper) or inhibition (lower) for each excited or inhibited neuron. Dots represent the median proportion of significantly excited or inhibited bins; lines represent the inter-quartile range. **p<0.01, Wilcoxon. (K) and (L) Same as (I and J), but for neurons contralateral to the injected hemisphere. There was no significant change in the proportion of excited or inhibited bins (p=0.20 and p=0.17, respectively, Wilcoxon).

DOI: https://doi.org/10.7554/eLife.34955.015

Second, we examined reward-associated firing in three free-fed populations: neurons ipsilateral to CTAP injection, neurons contralateral to CTAP injection, and neurons from uninjected subjects. Histograms aligned to receptacle entry show that subpopulations of neurons were excited and inhibited during reward consumption (*Figure 8*). These neural responses were not merely continuations

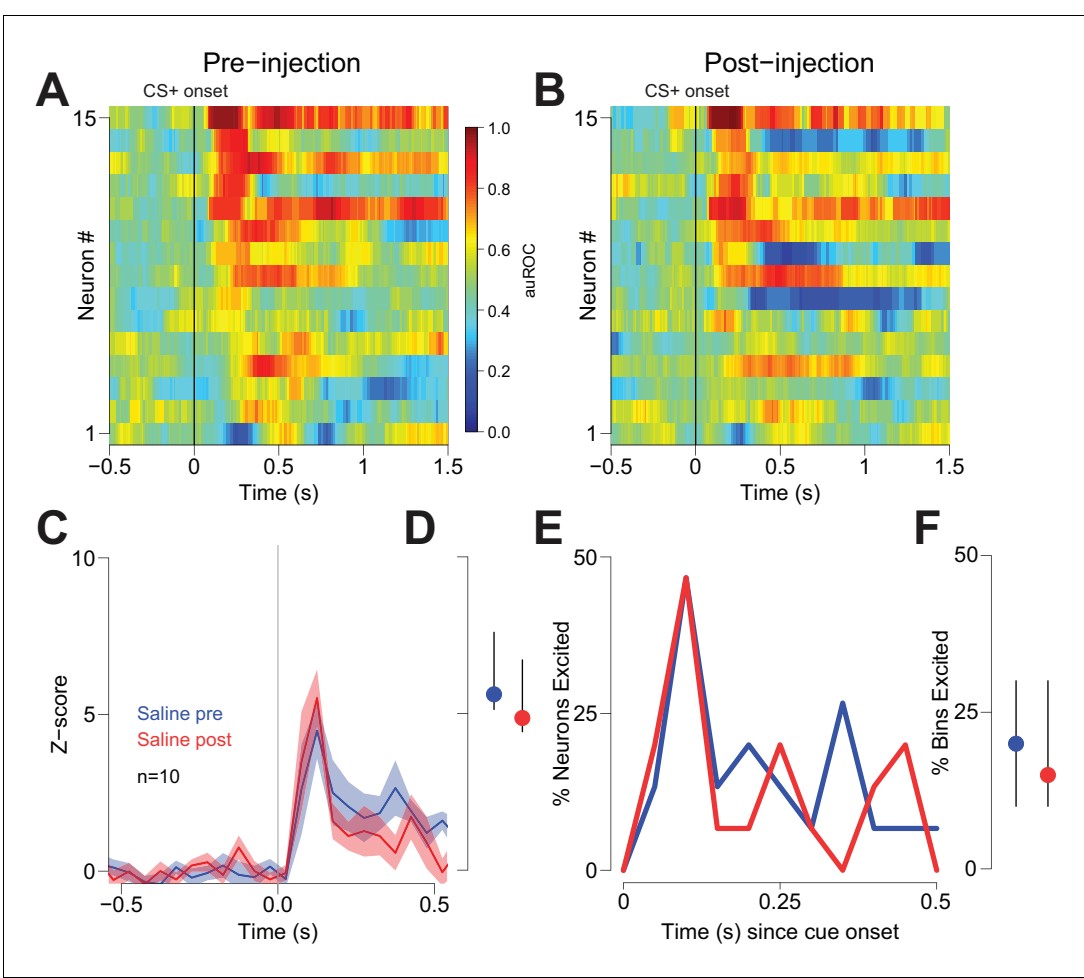

**Figure 7.** Vehicle injection does not affect cue-evoked excitations. (A) and (B) Population heat maps for neurons recorded from free-fed rats pre-vehicle injection and post-vehicle injection. (C) PSTH of neurons classified as cue-excited for pre- vs. post- injection epochs. (D) Dot plot indicating median and interquartile range of the response in each epoch (pre- vs post-injection) of cue-excited neurons between 100–300 ms following cue onset. (E) Proportions of cue-excited neurons in each 50 ms bin following cue onset pre-injection (blue trace) vs post-injection (red trace). (F) The proportion of bins with significant excitation for each cue-excited neuron pre- (blue trace) vs post-injection (red trace).

DOI: https://doi.org/10.7554/eLife.34955.016

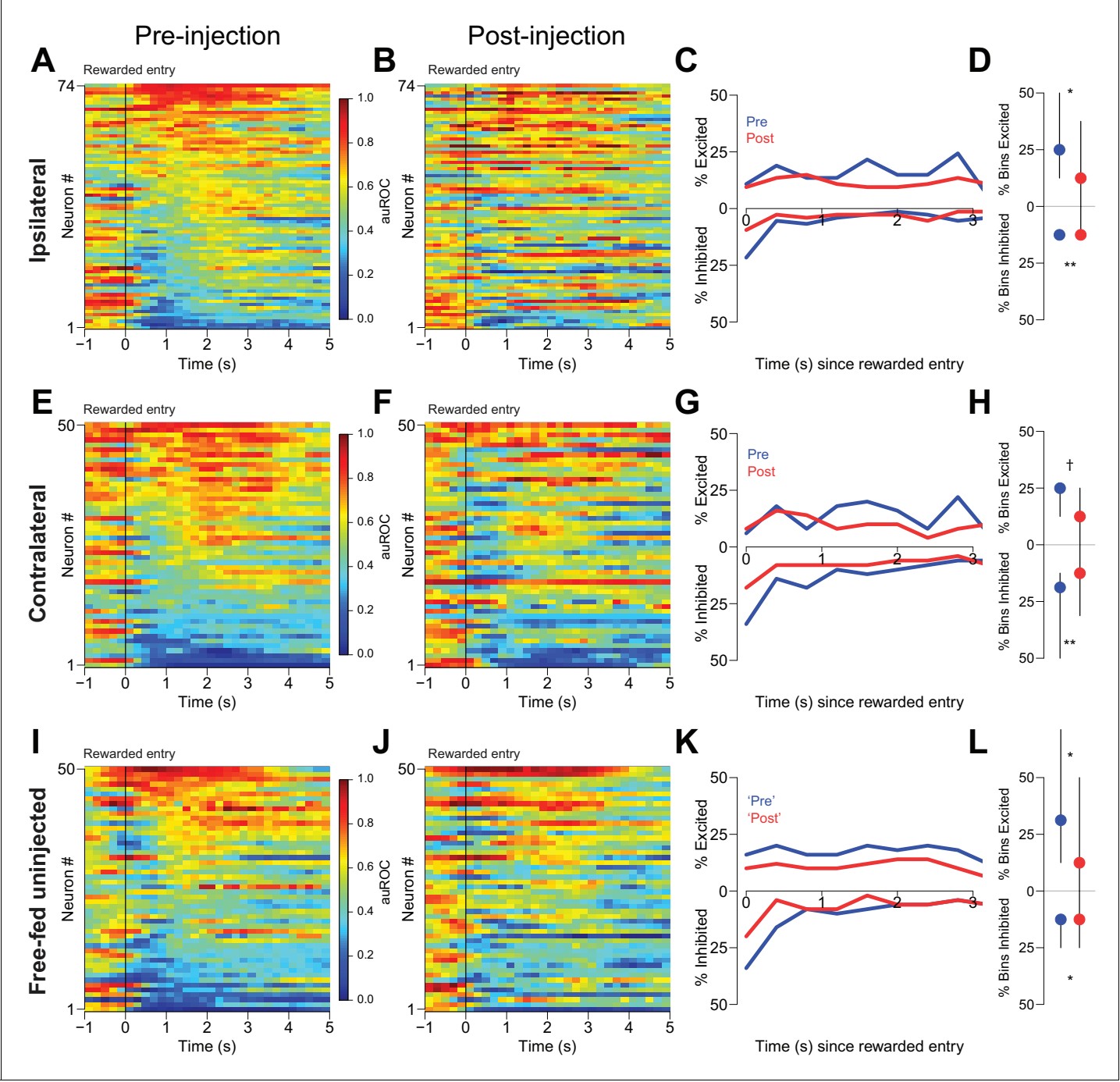

**Figure 8.** CTAP does not affect reward-associated firing. (A, E, I) Pre-injection neural activity aligned to the onset of rewarded receptacle entries. The baseline period is the pre-cue baseline for each rewarded trial in the respective epoch; each bin is 200 ms. Neuronal populations were recorded ipsilateral to CTAP injection (A), contralateral to CTAP (E) and in uninjected animals (I). (B, F, J) Post-injection neural activity for the same three populations. (C) Proportions of significantly excited (upper traces) and inhibited (lower traces) neurons in each 400 ms bin following rewarded receptacle entry pre-injection (blue traces) vs post-injection (red traces) for ipsilateral neurons. (D) The proportion of bins with significant excitation (upper) or inhibition (lower) for each significantly excited or inhibited neuron (see 'Analysis of neural data' in Materials and methods for inclusion criteria). Dots represent the median proportion of excited or inhibited bins; lines represent the inter-quartile range. (G, H) and (K, L) are the same as (C and D), except for contralateral neurons and free-fed non-injected neurons, respectively. For (I, J and K), only sessions for which there were at least six responded trials in each epoch were included in this analysis, as that was the minimum number of responded trials in either epoch of the unilateral sessions. †, p<0.10; *p<0.05; **p<0.01, Wilcoxon.

DOI: https://doi.org/10.7554/eLife.34955.017

*Figure 8 continued on next page*

*Figure 8 continued*

The following figure supplements are available for figure 8:

**Figure supplement 1.** Receptacle visit histograms following rewarded entries during unilateral injection sessions.
DOI: https://doi.org/10.7554/eLife.34955.018
**Figure supplement 2.** Excitations prior to rewarded receptacle entries mostly consist of sustained cue-evoked excitations that terminate upon receptacle entry.
DOI: https://doi.org/10.7554/eLife.34955.019

of cue-evoked excitations and inhibitions as cue-evoked activity tended to end prior to receptacle entry (*Figure 8—figure supplement 2*). Although we found significant post-injection decreases in reward-evoked excitations and inhibitions in both ipsilateral and contralateral populations (*Figure 8A–H*), we also observed similar changes in neurons from uninjected subjects simply by breaking up the responses into identical time epochs as the injected neurons (*Figure 8I–L*). Therefore, within-session changes in neural modulation to reward consumption cannot be attributed to the presence of CTAP. (Decreases in the magnitude of cue-evoked excitation across these epochs in uninjected animals were not observed; *Figure 8—figure supplement 1C–F*.) Finally, unilateral CTAP injection did not significantly affect either the total time spent in the receptacle during reward consumption or the overall number of receptacle entries during the consumption epoch (*Figure 8—figure supplement 1A,B*), indicating that the change in reward-associated firing observed during unilateral injection sessions is not a consequence of a change in consumption behavior. Taken together, these analyses indicate that in free-fed rats, declines in reward-associated firing over the course of the session are unlikely to be due to a MOR-dependent mechanism. In addition, they suggest that the CTAP-induced impairment of cued approach behavior (*Figures 1* and *4*) is very unlikely to result from changes in reward-associated firing.

## Discussion

In states of relatively high satiety, humans and animals greatly favor calorie-dense foods over less palatable options – a preference that likely contributes to overconsumption and obesity. Our results reveal a potential neural mechanism underlying this preference. We find that blockade of MORs in the NAc core attenuates both cue-evoked approach to high-fat food and the encoding of those cues by NAc neurons, and that these effects are observed only in rats fed ad libitum chow, and not in food-restricted (relatively hungry) rats. These effects could not be attributed to changes in consumption-related behavior or firing. Notably, NAc cue-evoked excitations are causal to cued approach (*du Hoffmann and Nicola, 2014*), suggesting that a novel and fundamentally important role of the NAc opioid system is to promote approach to highly palatable food specifically when there is no immediate homeostatic need for calorie intake. Together, these findings suggest NAc MORs as a target for development of treatments that limit overeating, consistent with the present use of drugs that block MORs as viable therapeutic options for the treatment of obesity (*Apovian, 2016*; *Ziauddeen et al., 2013*).

Although the NAc opioid system has long been implicated in the regulation of food intake (*Castro and Berridge, 2014*; *Kelley et al., 2005*; *Nicola, 2016*; *Peciña et al., 2006*; *Selleck and Baldo, 2017*), the MOR effects identified here are characterized by several features that were not necessarily predicted by previous studies. We find that activation of NAc MORs by endogenous ligands promotes appetitive behavior by increasing neural activity that drives approach to food, whereas NAc MORs appear to contribute little (if at all) to neural activity related to consumption. The latter conclusion appears to contrast with prior evidence that activation of these receptors by exogenous ligands increases hedonic taste reactions (*Peciña and Berridge, 2000*), which should be controlled by NAc neuronal activity occurring during consumption. However, because we targeted our electrodes to the NAc core, whereas hedonic taste reactions are promoted by MOR agonist injection in a very specific zone of the NAc shell (*Peciña and Berridge, 2000*), our results do not preclude the possibility that endogenous opioids promote taste reactivity (and perhaps hedonia) by influencing the consumption-related firing of NAc shell neurons.

On the other hand, injection of MOR agonists into either the NAc core or shell increases consumption of palatable food (*Bakshi and Kelley, 1993*; *Katsuura and Taha, 2014*; *Mucha and*

Iversen, 1986; Ward et al., 2006; Woolley et al., 2006; Zhang et al., 1998; Zhang and Kelley, 1997), which must be due to promotion of some form of NAc neuronal activity that drives consumption. Our results suggest that, at least in the core, this form of neural activity is the pre-movement firing of NAc neurons, which can be activated by cues (McGinty et al., 2013; Morrison et al., 2017) and which drives initiation of approach to calorie-dense food (du Hoffmann and Nicola, 2014). Further supporting this idea, activation of MORs in the NAc core by exogenous agonists increases consumption of a high-fat liquid in part by increasing the number of licking bouts (perhaps as a result of increasing the number of approaches to the lickometer) (Katsuura et al., 2011; Lardeux et al., 2015). Moreover, exogenous activation of NAc MORs promotes operant behavior for food reward (Zhang et al., 2003), and in fact is sufficient to increase operant responses to cues predictive of high-calorie food in a Pavlovian-instrumental transfer (PIT) test (Peciña and Berridge, 2013). The latter observation in particular supports the hypothesis that MOR activation directly promotes approach behavior without an intermediate effect on neural activity during consumption because the PIT test is performed in extinction. Further arguing against an intermediate effect on consumption, we observed previously that injection of CTAP into the NAc does not reduce consumption of a high-fat liquid in free fed rats (Lardeux et al., 2015). Finally, we find that unilateral infusion of CTAP into the NAc greatly reduces cue-evoked firing (Figure 6), an effect that could not have been due to reduced consumption or other performance deficit because cued approach performance was unaffected by unilateral infusions (Figure 4). Thus, we conclude that NAc core MORs act primarily to promote food seeking rather than consumption itself.

Our results do not, however, rule out the possibility that MORs contribute to some aspect of the consumption- or reinforcement-related firing of NAc neurons. Indeed, excitations in a small population of NAc neurons encode the value of liquid rewards during consumption (Taha and Fields, 2005); because we did not vary reward value, we may have missed an effect of CTAP on this form of encoding. However, more likely contributors to consumption are the large population of NAc neurons that are inhibited in proportion to the rate of licking during consumption (Taha and Fields, 2005). Together with findings that experimental silencing of NAc neurons drives consumption (Reynolds and Berridge, 2001; Stratford and Kelley, 1997) whereas consumption is interrupted by brief excitation of the NAc (Krause et al., 2010), these observations suggest that naturally occurring reductions in the firing of a population of NAc neurons (possibly containing D1 dopamine receptors and projecting to the lateral hypothalamus; O'Connor et al., 2015) drive consummatory behavior. Our observation that consumption-related inhibitions are not reduced by CTAP indicates that these inhibitions do not depend on MORs, consistent with previous findings that generalized inhibition of the NAc results in nonspecific increases in food intake whereas MOR activation results in preferential increases in consumption of calorie-dense, and especially high-fat, food (Katsuura and Taha, 2014; Ward et al., 2006; Woolley et al., 2006; Zhang et al., 1998; Zhang and Kelley, 1997). This specificity in the case of MOR agonists may be due in part to increased approach to calorie-dense food, perhaps via enhanced firing of NAc core neurons that drive such approach behaviors.

Strikingly, CTAP reduced cued approach when the reward was cream, but not liquid sucrose. The latter observation is consistent with previous studies indicating that activation of NAc MORs induces a preference for fat over carbohydrates (Taha, 2010; Zhang et al., 1998); however, similar manipulations also induce greater preference for the already-preferred flavor of two foods with equivalent nutritional content (Woolley et al., 2006). Although we cannot rule out the possibility that relatively greater preference for (or palatability of) cream vs sucrose was instrumental to the much greater effects of CTAP when the cue predicted cream as opposed to sucrose, the sucrose concentration (3%) was chosen such that the animals' behavior was similar to that observed with cream reward (Figure 1C,D), suggesting that it was the difference in nutrient content, not preference, that dictated the difference in dependence on NAc MORs. Although cream contains, in addition to fat, small quantities of certain nutrients that are absent from sucrose solution (e.g., protein, lactose), these minor components of cream are unlikely, on their own, to support appetitive approach and consumption in the free-fed state. Thus, our results suggest that NAc MORs specifically promote approach to high-fat foods. Even if flavor-based preference or palatability played a role, we note that preferred palatable foods tend to be calorie dense, supporting a role for NAc MORs in overconsumption that leads to obesity.

Such a role is further supported by the remarkable observation that CTAP affected neither cued approach behavior nor cue-evoked neural activity in food-restricted animals, despite markedly

reducing both in rats given ad libitum access to chow. To our knowledge, this is the first report of a satiety state-dependent contribution of endogenous NAc opioids to food-seeking behavior. Indeed, few studies have examined whether blockade of NAc MORs (as opposed to activation with exogenous agonists) impacts food-seeking. One exception is the observation that β-funaltrexamine (β-FNA), a long-lasting MOR antagonist, reduces rats' speed during runway approach to calorie-dense food (*Shin et al., 2010*). However, NAc injection of β-FNA has also been shown to reduce spontaneous locomotion (*Kelley et al., 1996*), suggesting that it may have had non-specific effects. Such effects could potentially also explain the reduction in calorie-dense food consumption after β-FNA injection in the NAc (*Bodnar et al., 1995*; *Kelley et al., 1996*; *Lenard et al., 2010*; *Shin et al., 2010*). Intriguingly, the MOR antagonist we used, CTAP, does not impair spontaneous locomotion when injected in the NAc (*Figure 4—figure supplement 1D*), and is also apparently less effective than β-FNA in reducing consumption (*Katsuura et al., 2011*; *Lardeux et al., 2015*) although a study in rabbits reports greater reductions in consumption (*Ward et al., 2006*). One possibility consistent with our results is that CTAP impairs approach to food as opposed to consumption itself; differences in CTAP effects on amount of freely available food consumed could be due to differences in experimental conditions such as size of the test chamber (and thus degree of approach required), species, nutrient content, and satiety state.

The stark difference in effects of CTAP in free fed and restricted animals raises three important topics for further research. The first is to determine the degree of restriction that is sufficient to eliminate the dependence of cued approach on NAc MORs. Although MOR antagonists can reduce food consumption after mild (<24 hr) restriction (*Bodnar et al., 1995*; *Kelley et al., 1996*), this may not be the case for cued approach behavior. If endogenous opioids in the NAc promote cued approach to fatty foods when restriction is much less severe than the chronic restriction used here, it would imply that this neural system contributes to caloric intake when meal patterns are more natural than the extremes employed here (freely available chow and severe restriction).

The second question is the mechanism whereby endogenous MOR ligands promote cued approach. As observed previously (*Ambroggi et al., 2011*; *du Hoffmann and Nicola, 2014*; *McGinty et al., 2013*; *Morrison et al., 2017*; *Nicola et al., 2004a*), we found that prominent populations of NAc neurons are excited and inhibited by cues that evoke approach behavior (*Figure 2A, B*). Previously, we established that these changes in firing begin prior to initiation of approach movement, and that the magnitude of the firing changes predicts the latency and speed of approach (*McGinty et al., 2013*; *Morrison et al., 2017*). Cue-evoked excitations, but not inhibitions, are dopamine-dependent; because injection of dopamine receptor antagonists reduces both cued approach and cue-evoked excitations (*du Hoffmann and Nicola, 2014*), the excitations are likely causal to the subsequent approach behavior. These observations suggest that activation of MORs by endogenous opioids could increase cue-evoked excitation via a direct action on NAc neurons, on the glutamatergic terminals that likely drive the excitation, or on inhibitory interneurons or inputs that limit the magnitude of cue-evoked excitation. Because MOR effects tend to be inhibitory, the latter hypothesis is most likely.

Alternatively, MOR activation by endogenous opioids could promote the release of dopamine, which could, in theory, promote greater cue-evoked firing and hence increase the probability of an approach response. This idea is consistent with previous findings that exogenous activation of MORs can increase dopamine levels (*Borg and Taylor, 1997*; *Hipólito et al., 2008*; *Hirose et al., 2005*; *Okutsu et al., 2006*; *Yoshida et al., 1999*), and with observations that dopamine release can be modulated at the terminal (*Cachope and Cheer, 2014*; *Wenzel and Cheer, 2018*). Moreover, dopamine neurons are strongly regulated by the state of caloric need, with greater activation in higher need states (*Meye and Adan, 2014*; *Nicola, 2016*), and in fact dopamine release evoked by food-predictive cues is greater in food-restricted rats than free-fed rats (*Aitken et al., 2016*; *Cone et al., 2014*) – an observation that could explain the present finding that NAc cue-evoked excitations are greater in restricted than free-fed animals (*Figure 2*). According to this hypothesis, dopamine levels in the free-fed state are insufficient to raise cue-evoked firing above the threshold for reliably obtaining a cue-evoked approach response. However, when the subject is in an environment in which calorie dense and/or high-fat food is available, neurons that release the endogenous agonist of NAc MORs in the NAc are activated to release the opioid, and the resulting activation of MORs raises the dopamine level such that the magnitude of cue-evoked firing is sufficient to evoke a behavioral response. In contrast, in food-restricted subjects, the dopamine level is so high that either further

increases are not possible, or the cue-evoked firing of NAc neurons is maximal such that further increases in dopamine are without effect.

The third question is the source and nature of the opioid peptides that activate MORs to promote food-seeking. Presumably, the endogenous ligand for NAc MORs is enkephalin released by the large population of D2 receptor-expressing spiny neurons (*Gerfen et al., 1990*; *Mansour et al., 1995*). The peptide could be released by the extensive axon collaterals of these neurons within the NAc; alternatively, while it has not been demonstrated in the NAc, opioids can be released somatodendritically and act as a retrograde messengers (*Iremonger and Bains, 2009*; *Wagner et al., 1993*; *Wamsteeker Cusulin et al., 2013*). The conditions under which enkephalin release in the NAc is increased are unknown; however, in the dorsomedial striatum, enkephalin levels are elevated during meal onset (*DiFeliceantonio et al., 2012*), suggesting that information about the availability of food drives release of the peptide. One possibility is that enkephalin release is tonically promoted when the subject is in an environment in which fatty foods are available; another is that release occurs precisely at cue onset in response to discrete cues that predict fat, but not carbohydrates. Further investigation of the hypothesis that NAc enkephalin release is regulated by fat availability, and the mechanisms by which this could occur, is clearly warranted.

The mechanism we propose here – that enkephalin levels are elevated by fat availability, and these high enkephalin levels promote dopamine release that in turn increases cue-evoked excitations that drive cued approach to high-fat food – is partially speculative, but it is fully consistent with the present and previous results and provides a starting point for further exploration. Importantly, our results indicate that the neural mechanisms underlying appetitive behavior must be considered when studying the contribution of opioids (and other neuromodulators) in the forebrain to food intake regulation. Cued approach is only one form of appetitive behavior, but our demonstration that endogenous opioids bias this form of behavior towards fat seeking suggests that opioids may have similar effects on the neural mechanisms that control other, more complex and/or more cognitive appetitive behaviors, such as deciding among simultaneously-available food options. Although MOR antagonists are currently used to treat obesity (*Apovian, 2016*; *Ziauddeen et al., 2013*), a more refined understanding of the impact of endogenous opioids on appetitive behaviors is required to understand how these drugs work, and to identify future targets for more specific and effective treatments that reduce preference for calorie-dense or high-fat foods.

## Materials and methods

### Animals

52 male Long-Evans weighing between 275 and 300 g were obtained from Charles River Laboratories and singly housed for a week before handling. Each rat was then handled for several minutes daily for 3 days to habituate them to the experimenter. Rats were randomly allocated to their experimental groups. Those designated for experiments requiring food restriction were limited to ~15 g of rodent chow per day for at least one week prior to the start of the experiment (to achieve 90% free-feeding weight), whereas free-fed animals had unlimited access to chow. All animals had unlimited access to water in their home cages. All procedures involving animals were in accordance with the National Institutes of Health Guide for the Care and Use of Laboratory Animals and were approved by the Institutional Animal Care and Use Committee at Albert Einstein College of Medicine.

### Operant chambers

Two styles of operant chambers were used in this study. For behavioral pharmacology experiments, chambers measured 30.5 cm x 24.1 cm and were supplied by Med Associates (St. Albans City, VT); chambers reserved for electrophysiology experiments measured 40 cm x 40 cm and were custom-designed. All chambers were outfitted with a reward receptacle equipped with an infrared head entry detector (Med Associates), as well as two 28 V house lights, a 65 dB white noise generator, and speakers for generating auditory cues. Reward was delivered via a syringe pump connected to the receptacle using 3/16" steel-reinforced PVC tubing to ensure consistent volume of reward delivery. All operant chamber hardware was controlled via custom-written Med-PC scripts.

## CS task and training

All rats used in this study were ad-libitum fed for the duration of training. The day before the first training session, rats were given access to heavy cream (per 100 g: 37 g fat, 2.8 g carbohydrate including 0.1 g sugars, 2.1 g protein) or 3% sucrose solution in their home cages to familiarize them with the reward. Training sessions lasted 2 hr. On the first day of training, rats were rewarded for simply entering the reward receptacle, with a 10 s timeout between rewarded entries. If they obtained >50 rewards, they advanced to the next phase of training; otherwise the current phase was repeated. In the second training phase, rats were presented with a reward-predictive CS+: either a siren tone (frequency cycle between 4 and 8 kHz over 400 ms) or an intermittent tone (6 kHz tone on for 40 ms, off for 50 ms) was played for a maximum duration of 5 s at a fixed intertrial interval (ITI) of 15 s. Head entries into the receptacle during presentation of the CS+ resulted in termination of the cue and delivery of ~50 µl heavy cream or 3% sucrose solution (although each rat received only one reward type). After rats obtained >50 rewards in a session, they were advanced to the full CS task, in which the CS+ or a neutral CS- (the siren tone for rats whose CS+ was the intermittent tone, and vice versa) were presented at ITIs randomly selected from a truncated exponential distribution (mean = 30 s, minimum of 10 s, maximum of 150 s). The CS- was presented for 5 s, regardless of receptacle entry, and had no programmed consequence. Rats were considered fully trained once they responded to >40% of CS+ presentations and had a discrimination index (defined as the number of CS+ responses divided by the total number of cue responses) of at least 0.67, indicating that rats reliably discriminated between the CS+ and CS-.

## Cannulated microelectrode arrays

Electrode arrays were custom-designed and assembled as previously described (*du Hoffmann et al., 2011*; *du Hoffmann and Nicola, 2014*). Briefly, each array consisted of 8 Teflon-insulated tungsten microwires (A-M Systems) encircling a 27-ga microinjection guide cannula. Each electrode was checked to ensure its impedance fell in the range of 90–110 MΩ. Electrodes and cannulae were mounted inside a drivable casing; a hex screw enabled the entire assembly to be driven along the dorsal-ventral axis of the NAc. Each full revolution of the screw drove the array ~350 µm. Once assembled, wires were soldered onto 10-pin connectors (Omnetics) and impedances were re-measured to ensure connection patency. A silver ground wire was soldered to the last pin on the connector.

## Surgeries

After rats reached criterion performance on the CS task, they were implanted either with custom-built bilateral cannulated microelectrode arrays aimed at the NAc core or with bilateral 26 ga micro-injection cannulae (Plastics One, Roanoke, VA) aimed at the NAc core as described previously (*du Hoffmann et al., 2011*; *du Hoffmann and Nicola, 2014*; *Nicola, 2010*). Rats were anesthetized with isoflurane (1–2%) and placed in a stereotaxic apparatus. From Bregma, cannulated arrays were implanted at AP +1.4 mm, ML ±1.5 mm, and DV −6.5 mm, while microinjection cannulae were implanted at AP +1.2 mm, ML ±2.0 mm, and DV −5.7 mm (microinjectors were designed to extend 2 mm beyond the cannulae tips, to a target of DV −7.7 mm). Implants were secured using dental acrylic bound to six screws fixed to the surface of the skull. Steel obdurators (Plastics One) were inserted into the cannulae to prevent them from clogging. For electrode surgeries, ground wires were inserted into the brain at a posterior location, and connectors were fixed to the implant at the posterior aspect of the cap. Antibiotics (Baytril) were provided immediately before and 24 hr after surgery. Rats were allowed one week to recover from surgery before re-training commenced.

## Microinjection experiments

After recovering from cannulation surgery, a subset of rats were food-restricted for one week. Food restriction was concomitant with re-training. All other rats continued to have ad-libitum access to food. After behavioral responding was re-established, we began the microinjection procedures. Microinjectors (33 ga, Plastics One) were affixed to polyethylene tubing that was back-filled with mineral oil and connected to two 1 µl Hamilton syringes which were under the control of a microinjection pump (KD Scientific, Holliston, MA). On the first day, rats received a mock injection to habituate them to the injection procedure. Rats were gently restrained while microinjectors were inserted

into the guide cannulae and left in place for 1 min prior to the start of the infusion to allow the tissue to equilibrate around the injectors. D-Phe-Cys-Tyr-D-Trp-Arg-Thr-Pen-Thr-NH2 (CTAP) (Sigma-Aldrich; 0, 2, or 4 μg/side), was dissolved in 0.9% saline and infused at a rate of 0.25 μl/min for 2 min for a total infusion volume of 0.5 μl per hemisphere. Post-infusion, injectors were left in the cannulae for 1 min post-infusion to allow the drug to diffuse into the tissue. After injection, rats were immediately placed in the operant chambers and the behavioral session was started. The order in which each rat received each drug dose was pseudo-randomized across injection days. Injection days were interleaved with non-injection days to ensure that rats' behavior returned to baseline performance levels.

### Recording/Injection experiments

Following recovery from cannulated electrode array implantation, a subset of rats were food-restricted as in the microinjection experiments. After consistent behavior was re-established, rats were tethered to a 16-channel commutator by the recording cable, which allowed for free rotational movement of the animal during neural recordings. On simultaneous recording/injection days, 33-ga microinjectors were affixed to polyethylene tubing pre-filled with mineral oil and connected to a 2-channel fluid swivel to allow for free rotational movement, terminating at a microinjection pump (KD Scientific) that sat atop the chamber. Drug was then loaded into the microinjector tips such that the interface between saline-dissolved drug and mineral oil was visible; the location of this interface was marked on the fluid lines. Prior to the start of the session, rats were tethered to the recording apparatus via the recording cable and either one (for unilateral injections) or two (for bilateral injections) microinjectors targeting a depth of 500 μm beyond the electrode tips were inserted into the guide cannula and taped to the recording cable such that they could not be readily removed by the rat. Once secured, fluid lines were visually inspected to ensure that the drug-oil interface remained at the marking on the fluid line, assuring that drug had not leaked out prematurely. Neural signals were then examined online to isolate active channels (see Materials and methods section *Acquisition of neural data*) and the behavioral session commenced. To obtain a neural and behavioral baseline, rats performed the task for 2000 s (~33 min), at which point the drug pump was remotely triggered, initiating the infusion of a 0.5 μl volume of either saline or 8 μg/side CTAP over a period of 12 min. This procedure allowed us to compare behavior and neural activity during the baseline window to behavior and neural activity during an equivalent-duration post-injection window. The higher 8 μg/side CTAP dose was chosen to mitigate the possibility that potentially partial drug effects using lower doses would mask changes in behavior and neural activity in the briefer window of examination (~33 min) used in these experiments.

### Acquisition of neural data

Rats were connected to a recording cable outfitted with a 16-channel headstage. The cable was connected to a multichannel commutator that was in turn connected to a pre-amplifier, where the neural signals were amplified by 2,000–20,000X and band-pass filtered at 250 Hz and 8.8 kHz before being passed to a 40 kHz multi-unit acquisition processor. Prior to the start of a session, each channel was examined for putative unit activity using SortClient (Plexon Inc, Dallas, TX) and optimized for gain and threshold.

### Analysis of neural data

Following acquisition, putative units were isolated manually using Offline Sorter (Plexon). To be included in the analysis, units had to have an absolute amplitude $\geq$75 μV and $\leq$0.1% of all inter-spike intervals could be $\leq$2 ms. When multiple units were recorded on the same channel, cross-correlograms were used to ensure that spikes were assigned to the appropriate unit and that the units were well-isolated from one another. If these conditions were not met, then the spiking activity on these ambiguous channels was discarded. Spike timestamps were then imported into R, combined with the associated behavioral data, and analyzed using custom routines ([*Caref, 2018*]; https://github.com/kcaref/neural-analysis; copy archived at https://github.com/elifesciences-publications/neural-analysis). Neurons were classified as cue-excited if they exceeded the 99.9% confidence interval of a Poisson distribution comprised of a 10 s pre-cue baseline for at least one 50 ms bin following CS+ onset and up to 500 ms post-cue onset. Neurons were classified as cue-inhibited if they fell

below the 99% confidence interval for at least one 50 ms bin. A less stringent detection threshold was used for inhibitions because many NAc neurons exhibit low baseline firing rates, making it harder to detect inhibitions due to floor effects. Neurons were classified as significantly excited during reward consumption if firing in at least one 400 ms bin following the rewarded receptacle entry exceeded the 99% confidence interval of a Poisson distribution comprised of a 10 s pre-cue baseline; they were classified as consumption-inhibited if firing fell below the 99% confidence interval. For simultaneous recording/injection experiments, classification of neural responses was performed only during the pre-injection epoch so that any potential drug effects would not contribute to the neuronal classification.

To construct heat maps illustrating the frequency and magnitude of cue-evoked excitations and inhibitions, for each neuron a receiver operating characteristic (ROC) curve was computed in 10 ms bins from 1 s prior to cue onset to 1.5 s after. The ROC curve used data from each trial to compare the firing rate in each bin to the 1 s baseline. The area under the ROC curve (auROC) for each bin was then displayed as the smoothed mean of a 200 ms sliding window. To construct heat maps for consumption-evoked activity, an auROC value was computed for each 200 ms bin from 1 s before the rewarded receptacle entry to 5 s after the rewarded entry using the pre-cue epoch as the baseline. auROC values are displayed as the smoothed mean of an 800 ms sliding window. An auROC value of 1 corresponds to very strong excitation; a value of 0 corresponds to very strong inhibition, and a value of 0.5 indicates no change in evoked firing relative to baseline. Because auROC values are always between 0 and 1, these values can be used to visually compare cue-evoked firing across different neuronal populations and conditions. All statistical comparisons were performed on non-smoothed data.

## Analysis of video tracking data

When possible, the rat's position was tracked by an overhead camera at 30 fps using 2 LEDs mounted on the neural recording headstage. Video tracking was conducted using the CinePlex software suite (Plexon). Tracking data were preprocessed as described previously (*McGinty et al., 2013*). Briefly, the locomotor index (LI) was computed for each frame by taking the standard deviation of the frame-to-frame difference in x-y position for four preceding and four succeeding video frames. Thus, the LI for each frame is a smoothed spatial and temporal representation of the rat's speed over nine frames (~300 ms). The resulting distribution of LIs for all video frames was then fitted with a double-Gaussian function; the subject was considered still when LI values were below the threshold between Gaussian peaks, and moving if the LI value was above this threshold. The LI threshold value differed from rat to rat and depended in part on the rat's overall activity during the session. Nearest neighbor analysis was then conducted to determine the video frame corresponding to the start of behavioral and task events such as CS+ onsets. The latency to initiate locomotion following cue onset was computed by subtracting the timestamp of the cue from the timestamp of the first frame following cue onset whose locomotor index exceeded the threshold value.

## Statistical analyses

For all experiments, results were considered statistically significant if p<0.05. For behavioral pharmacology experiments, drug effects were evaluated using two-way ANOVA. When appropriate, post-hoc tests were conducted using the Holm-Sidak p-value adjustment for multiple comparisons. For comparisons of event-evoked firing in separate neural populations, Wilcoxon rank sum tests were used; for within-session comparisons of the same neural population, Wilcoxon signed-rank tests were used. To compare pre- vs. post-injection baseline firing rates, a 95% confidence interval was constructed around the slope of the regression line resulting from plotting the pre-injection baseline against the post-injection baseline. If the confidence interval included 1, the result was not statistically significant.

## Generalized Linear Model (GLM) fitting

For the modeling procedures employed in *Figure 2*, the contribution of behavioral and spatial parameters to cue-evoked firing was examined using a GLM,

$$\ln(Y) = \beta_0 + \beta_1 x_1 + \beta_2 x_2 + \varepsilon, \tag{1}$$

where Y is the number of spikes in the window between 50 and 500 ms post-cue, $\beta_{0...n}$ are the regression coefficients for each dependent variable, $x_{1...n}$ are the values of the dependent variables (i.e., the regressors such as distance, speed, etc.), and $\varepsilon$ is an error term. This form of the GLM assumes Poisson-distributed values of Y, and as such the log transformation of Y is in reference to the model fit, not the actual data. Each population of cue-excited neurons (i.e., neurons from free-fed rats and those from food-restricted rats) was pooled to facilitate population-level comparisons. To both validate GLM fits and to evaluate whether two population GLMs model the same overall population, we used the following procedure,

$$F = \frac{\frac{SS_t - SS_p}{(m+1)(k-1)}}{\frac{SS_p}{DF_p}} \tag{2}$$

where $SS_t$ is the total residual sum of squares from a GLM fitted to the combined data, $SS_p$ is the pooled residual sum of squares from each individual GLM, $m$ is the number of regressors in each individual GLM, $k$ is the number of GLMs being evaluated, and $DF_p$ is the pooled residual degrees of freedom from each individual GLM (Zar, 1999). The resulting F statistic is then converted to a p-value; the numerator degrees of freedom is the $(m+1)(k-1)$ expression and the denominator degrees of freedom is $DF_p$. If two GLMs model the same overall population, then the resulting p-value will be > 0.05. To assess GLM fit, we employed a within-population bootstrapping approach. For each neural population, we fitted a GLM using a randomly-sampled selection of 50% of CS+ trials on which animals responded to the cue. We then fitted a second GLM using the remaining trials, and computed a p-value using *Equation 2*. This procedure was performed 1000 times for each neural population, with the reasoning that if the model fits are adequate, then p will be > 0.05 on 95% of the bootstrapped permutations.

## Histology

Before sacrifice, all rats were injected with Euthasol (39 mg/kg pentobarbital) to induce deep anesthesia. Rats from microinjection experiments were decapitated and their brains were removed and stored in 4% paraformaldehyde solution. Rats from electrophysiology experiments underwent intra-cardiac perfusion of saline followed by 4% paraformaldehyde solution. They were then decapitated, and their brains were removed and stored in 4% paraformaldehyde solution. All brains were then sectioned into 50 μm slices using a vibratome. The slices were mounted on slides and cresyl violet-stained to facilitate examination of injection sites and cannula placements. One food-restricted rat from the microinjection experiment (*Figure 1*) died before its brain could be extracted and examined, but because the rest of its cohorts' cannulae were placed accurately, we decided to include this data in the study.

## Acknowledgements

This work was supported by NIH grants DA019473, DA038412, MH092757, and DA041725 to SMN; and by grants from the Klarman Family Foundation and NARSAD to SMN. We thank J Kim for expert technical assistance, R Coen-Cagli for helpful discussions, and D Robinson, S Morrison, J du Hoffmann, M Kazmierczak, M Vega-Villar and C Ward for comments on the manuscript.

## Additional information

### Funding

| Funder | Grant reference number | Author |
| --- | --- | --- |
| National Institutes of Health | DA019473 | Saleem M Nicola |
| Klarman Family Foundation | Pilot Award | Saleem M Nicola |
| Brain and Behavior Research Foundation | NARSAD Young Investigator Awards | Saleem M Nicola |
| National Institutes of Health | DA038412 | Saleem M Nicola |

| National Institutes of Health | MH092757 | Saleem M Nicola |
| National Institutes of Health | DA041725 | Saleem M Nicola |
| Klarman Family Foundation | Two Year Award | Saleem M Nicola |

The funders had no role in study design, data collection and interpretation, or the decision to submit the work for publication.

## Author contributions

Kevin Caref, Data curation, Software, Formal analysis, Validation, Investigation, Visualization, Methodology, Writing—original draft, Writing—review and editing; Saleem M Nicola, Conceptualization, Supervision, Funding acquisition, Methodology, Project administration, Writing—review and editing

## Author ORCIDs

Kevin Caref ⓘ http://orcid.org/0000-0001-6424-4272
Saleem M Nicola ⓘ http://orcid.org/0000-0001-9582-6312

## Ethics

Animal experimentation: All procedures involving animals were in accordance with the National Institutes of Health Guide for the Care and Use of Laboratory Animals and were approved by the Institutional Animal Care and Use Committee at Albert Einstein College of Medicine (protocols 20100103, 20130204, and 20160206).

## Decision letter and Author response

Decision letter https://doi.org/10.7554/eLife.34955.024
Author response https://doi.org/10.7554/eLife.34955.025

## Additional files

### Supplementary files

• Transparent reporting form
DOI: https://doi.org/10.7554/eLife.34955.020

### Major datasets

The following dataset was generated:

| Author(s) | Year | Dataset title | Dataset URL | Database, license, and accessibility information |
|---|---|---|---|---|
| Kevin Caref | 2018 | Data from: Endogenous opioids in the nucleus accumbens promote approach to high-fat food in the absence of caloric need | https://doi.org/10.5061/dryad.t8v2gg7 | Available at Dryad Digital Repository under a CC0 Public Domain Dedication |

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
