## [Decision Letter]

Thank you for submitting your article "Endogenous opioids in the nucleus accumbens promote approach to high-fat food in the absence of caloric need" for consideration by *eLife*. Your article has been favorably evaluated by a Senior Editor and three reviewers, one of whom, Richard D Palmiter (Reviewer #1), is a member of our Board of Reviewing Editors.. The following individual involved in review of your submission has also agreed to reveal their identity: Roger Adan (Reviewer #3).

The reviewers have discussed the reviews with one another and the Reviewing Editor has drafted this decision to help you prepare a revised submission.

*Summary:*

This is a tightly constructed paper showing that opioid action in the nucleus accumbens core (NAc) promote approach to calorie-dense foods during periods of satiety. By combining behavioral measures, infusion of an opioid antagonist and electrophysiological recording, the authors were able to demonstrate that opioids primarily affect cued approach behavior rather than consumption. Importantly, the opioid system had little effect when rats were hungry. Thus, the authors conclude that opioid action in the NAC contributes to non-homeostatic feeding, which likely contributes to obesity.

*Essential revisions:*

(Points 1, 3, 4, 6, 7 are essential; points 2 and 5 would be nice additions if data are available.)

1) Justify use of different doses of CTAP for different experiments.

2) Consider adding an experiment in which authors investigate the effect of CTAP when mice go from restricted feeding back to ad-lib feeding.

3) Discuss possible sources and timing of the opioid receptor agonist release and the implications.

4) Provide more details regarding whether the same neurons that respond during cue presentation also respond during reward delivery/consumption.

5) Adding data for one night of free feeding of the restricted rats would be nice.

6) Comment on why the yield of recorded neurons is different for ad lib and restricted groups of rats.

7) Is there a ceiling effect for feeding by the restricted rats such that opioids no longer have an effect?

See entire reviews below for more details on essential revisions.

*Reviewer #1:*

This is a tightly constructed paper showing that opioid action in the nucleus accumbens core (NAc) promote approach to calorie-dense foods during periods of satiety. By combining behavioral measures, infusion of an opioid antagonist and electrophysiological recording, the authors were able to demonstrate that opioids primarily affect cued approach behavior rather than consumption. Importantly, the opioid system had little effect when rats were hungry. Thus, the authors conclude that opioid action in the NAC contributes to non-homeostatic feeding, which likely contributes to obesity.

This study is well designed, the experiments are described clearly and the results discussed in a cogent manner. I have no major concerns.

*Reviewer #2:*

Caref and Nicola seek to determine the role of endogenous NAc opioids in reward-directed behavior and reward encoding. This work is built on previous findings that Mu opioid receptor activation in the NAc promotes food intake and intake of high fat foods in particular. Much of this work has focused on the delivery of Mu agonists. Here the authors utilize a Mu receptor antagonist to probe contributions from endogenous Mu ligand activity. They utilize a cue-induced approach task and find that, only in ad libitum fed rats, Mu receptor activation is required for cue-evoked behavior and NAc excitations. The latter finding is consistent with previous work from Nicola supporting that cue-evoked activations of NAc neurons are critical for conditioned approach. The authors model their data to demonstrate that the modulation of cue evoked NAc excitations by hunger cannot be attributed to differences in approach latency or motor behavior. Moreover, the dependency of NAc cue-evoked excitations on Mu receptors is not a secondary effect due to alterations in palatability. The work is elegant, the analyses are well-performed and the manuscript makes for an important contribution to our understanding of cue-driven feeding on high fat foods in the absence of caloric need. Thus, the findings enhance our understanding of neural processes that contribute to the obesity epidemic. I have the following questions that hopefully can be addressed to strengthen the manuscript:

1) There is a marked difference in the dose of CTAP used in behavioral experiments (2 or 4ug) versus those experiments that included electrophysiological recordings (8ug). No rationale for dose choice is given nor is there an explanation is given for the much higher dose used during the recordings. Do dose values reflect what was delivered to each side?

2) Given the striking differences between fed and restricted states, do the authors have any data from restricted rats that returned to ad libitum feeding? A demonstration of 'toggling on' the effect of CTAP in rats that were previously food restricted would be very powerful. This could be done at the strictly behavioral level.

3) In the Discussion, the authors speculate that MOR ligand may be sourced from NAc neurons themselves. Are the authors suggesting that the initiation of cue-evoked excitations could lead to enkephalin release that would then, in some feed-forward manner, enhance the ongoing response to the cue. Indeed, some of the data reflect fewer bins as time proceeds from cue onset that were statistically elevated following CTAP administration. Or do the authors speculate that MOR receptor activation is occurring in a less time-restricted manner? As there are no effects on baseline firing rate, the latter idea seems unlikely. Some deeper discussion of the source of MOR ligand and timing of MOR receptor activation is warranted.

4) Concerning the populations of neurons that responded during reward consumption, it is unclear if these are neurons with sustained responses from cue onset or whether some of these neurons were only modulated during reward consumption. In the "Analysis of neural data" section of the Materials and methods, only details regarding cue responses are described. If there are some neurons that respond exclusively during consumption, then modulation of this population by CTAP could potentially contribute to reinforcement. This effect could be masked by the population with ongoing heightened activity from the cue period. Regardless, more detail concerning the identification of consumption-related neurons and their relationship to cue-related neurons is warranted.

5) Food restriction appears to result in several effects on MSN activity: increases the proportion of neurons responding with an excitation, increases the magnitude of excitations, increases the duration of excitations (Figure 2C-F). However, the data from free-fed pre-injection rats demonstrates a relatively higher percentage of neurons responding with excitations and a percentage that is not different from restricted, re-injected rats. Thus, the proportion of neurons exhibiting excitations may not be a reliable predictor of free versus fasted state. Again, it would be interesting to see the effects of one night of free-feeding would be on these measures in restricted rats.

*Reviewer #3:*

In their article, Caref and Nicola describe an interesting and physiologically important observation that the MOR antagonist CTAP reduced CS responsivity in a food- and state-dependent manner, by means of in vivo pharmacology and single unit recordings. The authors should be applauded for the extensive analysis of their ephys experiment, and the many control experiments that were included. We do, however, have one major concern that we think should be addressed, which applies to some of their ephys findings.

In their Results, in the first paragraph of the subsection “Neural encoding of reward-predictive cues by NAc neurons is different in free-fed and food-restricted rats”, they say that in the ad lib condition, 83 neurons were recorded from 12 rats, whereas in the food restricted condition, 122 neurons were recorded from 5 rats. This is a very big difference in yield between these condition; ~7 neurons/animal ad lib, versus ~24 neurons/animal during food restriction. We suspect that the baseline firing rate of NAcc neurons increases due to the food restriction, which may have resulted in a selection bias of neurons in the ad lib condition. As such, recorded neurons in the food restricted condition will consist of a higher variety of neurons than in the ad lib fed condition, in which potentially only from a subpopulation of highly active neurons was recorded.

The authors should clarify why the yield per animal is so different between the conditions, and otherwise experimentally or analytically show that the population of neurons in the ad lib condition are similar to those recorded from in the food restricted condition. If this is not the case, the comparison that is made between neural activity between the two conditions (e.g. in Figure 2) should be interpreted with caution.

Related to this there may be a ceiling effect of increased responding in food restricted rats such that motivation in this task is so high that opioids do not modulate it.

---

## [Author Response]

We thank Dr Palmiter for his supportive comments.

Reviewer #2:

[…] 1) There is a marked difference in the dose of CTAP used in behavioral experiments (2 or 4ug) versus those experiments that included electrophysiological recordings (8ug). No rationale for dose choice is given nor is there an explanation is given for the much higher dose used during the recordings. Do dose values reflect what was delivered to each side?

We thank reviewer #2 for the supportive comments. While we show in Figure 1C that the 4 µg/side dose significantly attenuates responding to the CS+, it does not block it completely, particularly early in the behavioral session (i.e., immediately following the drug injection). Thus, when performing CTAP injections during our electrophysiology experiments, we were concerned that the drug effect might be masked in the ~33-minute post-injection window. To mitigate this possibility, we used a higher dose (8 µg/side), which is in line with doses used previously in the literature (5 – 8 µg) (Lardeux, Kim, and Nicola, 2015; Perry and McNally, 2013; Ward, Nicklous, Aloyo, and Simansky, 2006). Moreover, because unilateral injections have no effect on behavior, the use of a lower dose that less consistently impacts behavior when injected bilaterally would have made it more difficult to be certain that any given injection is biologically effective (i.e., it would have had a behavioral effect had both sides been injected). In contrast, bilateral injections at the 8 µg dose strongly and consistently attenuated behavioral responding, and so we can be certain that injections of this higher dose were biologically effective.

We have amended the Materials and methods section to include this justification, and we have clarified that these doses reflect the amount of drug delivered to each hemisphere.

2) Given the striking differences between fed and restricted states, do the authors have any data from restricted rats that returned to ad libitum feeding? A demonstration of 'toggling on' the effect of CTAP in rats that were previously food restricted would be very powerful. This could be done at the strictly behavioral level.

We agree with reviewer #2 that the demonstration of a toggle in the requirement of NAc MOR activation for cue responding would represent a powerful insight into the underlying mechanism. However, this experiment is tied up with the broader questions of time course and degree of restriction: how long must animals be food restricted before responding becomes CTAP-insensitive? (Hours? Days? An entire week, as used here?) How long must previously restricted rats be ad libitum-fed before responding becomes CTAP-sensitive? How much restriction (90% of free feeding weight? 95%?) is required for responding to become CTAP-insensitive? These are important questions to address in further studies, and the answers are necessary to design a “toggle” experiment. Because performing these experiments would be a major undertaking, we feel that it is best saved for a separate study.

3) In the Discussion, the authors speculate that MOR ligand may be sourced from NAc neurons themselves. Are the authors suggesting that the initiation of cue-evoked excitations could lead to enkephalin release that would then, in some feed-forward manner, enhance the ongoing response to the cue. Indeed, some of the data reflect fewer bins as time proceeds from cue onset that were statistically elevated following CTAP administration. Or do the authors speculate that MOR receptor activation is occurring in a less time-restricted manner? As there are no effects on baseline firing rate, the latter idea seems unlikely. Some deeper discussion of the source of MOR ligand and timing of MOR receptor activation is warranted.

As suggested, we have included a deeper discussion of the source of NAc enkephalin and the timing with which it carries out its effects on cue responding and cue-evoked excitation (next-to-last paragraph of the Discussion). Clearly the nature of the MOR ligand and the regulation of its release are important questions, but as there is very little data on which to base hypotheses, we have listed some possibilities without committing to any of them.

4) Concerning the populations of neurons that responded during reward consumption, it is unclear if these are neurons with sustained responses from cue onset or whether some of these neurons were only modulated during reward consumption. In the "Analysis of neural data" section of the Materials and methods, only details regarding cue responses are described. If there are some neurons that respond exclusively during consumption, then modulation of this population by CTAP could potentially contribute to reinforcement. This effect could be masked by the population with ongoing heightened activity from the cue period. Regardless, more detail concerning the identification of consumption-related neurons and their relationship to cue-related neurons is warranted.

The reviewer raises the plausible possibility that some of the consumption-modulated firing might be due to residual cue-evoked firing, and thus may be differentially affected by CTAP than firing that is exclusively consumption-modulated. To address this, we identified and excluded from analysis neurons that met *all* of the following criteria:

1) Neuron exhibited significant cue-evoked excitation as per our stated criteria;

2) Neuron was significantly excited in the 400-ms bin prior to rewarded receptacle entry;

3) Neuron was significantly excited in the first 400-ms bin following rewarded entry.

We defined significant excitation in the 400-ms bins as firing that exceeded the 99% confidence interval of a Poisson distribution comprised of 10 s of pre-cue baseline activity; this description has been added to the Materials and methods section under “Analysis of neural data”.

In fact, very few neurons met all 3 of these criteria for either excitations or inhibitions for each of the neural populations examined in Figure 8, suggesting that excitations during consumption are rarely continuations of excitations begun at cue onset. These “continuation” neurons are summarized in the following table:

Type of residual activityIpsilateralContralateralUninjectedExcitation536Inhibition021

Excluding the “continuation” neurons from the analysis did not statistically affect our results; therefore, we left them in.

To further investigate the possibility that consumption-related activity was the result of continued cue-evoked excitation, we focused on neurons that exhibited excitation just prior to receptacle entry by sorting the heat map histograms of firing aligned to receptacle entry by the magnitude of firing rate increase in the 1 sec prior to entry (relative to the pre-cue baseline; see heat maps on the left side of Figure 8—figure supplement 2, which we added in this revision of the paper). We found that these excitations tended to terminate prior to the animal entering the reward receptacle, and therefore these pre-entry excitations are not responsible for consumption-related excitations (which are observed, typically, in other neurons). Next, we sorted the entry-aligned histograms by the magnitude of excitation just after cue onset (between 100-300 ms), and found that those neurons that had the strongest post-cue excitations (i.e., those at the top of the heat maps on the right side of Figure 8—figure supplement 2) tended to also show pre-receptacle entry excitations, which terminated just prior to receptacle entry.

Together, these observations confirm that although cue-evoked excitations can often last until receptacle entry, excitations during reward consumption are rarely, if ever, the result of prolonged cue-evoked excitations, and instead are found in different neurons.

5) Food restriction appears to result in several effects on MSN activity: increases the proportion of neurons responding with an excitation, increases the magnitude of excitations, increases the duration of excitations (Figure 2C-F). However, the data from free-fed pre-injection rats demonstrates a relatively higher percentage of neurons responding with excitations and a percentage that is not different from restricted, re-injected rats. Thus, the proportion of neurons exhibiting excitations may not be a reliable predictor of free versus fasted state. Again, it would be interesting to see the effects of one night of free-feeding would be on these measures in restricted rats.

The reviewer is correct that in bilaterally-injected free-fed rats, the proportion of cue-excited neurons is higher than in the larger population of neurons recorded from free-fed rats that were not injected with drug. This is one reason why we did not make statistical comparisons of the proportions of cue-excited neurons in free-fed and food-restricted populations. We grant that with more recordings in each population, a valid statistical trend might emerge, but we are uncomfortable with stating such a result given this discrepancy (which may in part be a consequence of the lower number of recorded neurons for the bilaterally-injected population).

Additionally, we agree with reviewer #2 that a one-night free-feeding of food-restricted rats could provide valuable information on the effects of food restriction on NAc neuronal activity. However, the question of a single night’s free-feeding effects is part of the larger question, raised above in the response to point #2, about the time courses of restriction and ad libitum feeding with respect to CTAP effects, and we feel that it would be best answered in the context of a more thorough examination.

Reviewer #3:

[…] The authors should clarify why the yield per animal is so different between the conditions, and otherwise experimentally or analytically show that the population of neurons in the ad lib condition are similar to those recorded from in the food restricted condition. If this is not the case, the comparison that is made between neural activity between the two conditions (e.g. in Figure 2) should be interpreted with caution.Related to this there may be a ceiling effect of increased responding in food restricted rats such that motivation in this task is so high that opioids do not modulate it.

We thank Dr Adan for his supportive comments. To address whether a difference in baseline firing rate might underlie the discrepancy between the number of neurons recorded per animal in free-fed vs. food-restricted populations, we performed the additional analysis shown in Figure 2—figure supplement 2. This figure shows the superimposed distributions of baseline firing rates for both free-fed and food-restricted rats. As is evident from the histograms, the free-fed histogram overlaps cleanly with the food-restricted histogram. The colored arrows in the figure represent the median baseline firing rate of each population, and these are nearly identical. Unsurprisingly, a Wilcoxon rank sum test revealed no statistical difference between the two populations. Thus, if a difference in baseline firing rate underlies the observed discrepancy in neurons per animal, then that would imply the existence of a separate population of completely quiescent neurons in free-fed rats. This seems unlikely to us given that the recorded neurons with the lowest baseline firing rates were all derived from food-restricted rats (see the left-most bar in the figure).

We think the best explanation for the discrepancy is the unpredictable nature of in vivo electrophysiological recordings. Neuronal yield simply varies greatly across implants, for reasons that are not clear.